# The Unfolded Protein Response—Novel Mechanisms, Challenges, and Key Considerations for Therapeutic Intervention

**DOI:** 10.3390/cancers17223639

**Published:** 2025-11-13

**Authors:** P. M. Quan Mai, Tam-Anh Truong, Sai Kumar Samala, Bhoomika Muruvekere Lakshmisha, Prapannajeet Biswal, Khadijeh Koushki, Prudhvi Chand Mallepaddi, Geraldine Vijay, Sunil Krishnan

**Affiliations:** 1Vivian L. Smith Department of Neurosurgery, The University of Texas Health Science Center at Houston, Houston, TX 77030, USA; phuoc.minh.quan.mai@uth.tmc.edu (P.M.Q.M.); sai.kumar.samala@uth.tmc.edu (S.K.S.); bhoomika.muruvekerelakshmisha@uth.tmc.edu (B.M.L.); prapannajeet.biswal@uth.tmc.edu (P.B.); khadijeh.koushki@uth.tmc.edu (K.K.); prudhvi.c.mallepaddi@uth.tmc.edu (P.C.M.); geraldine.v.raja@uth.tmc.edu (G.V.); 2Department of Neuroscience, Johns Hopkins University, Baltimore, MD 21218, USA; ttruon23@jh.edu

**Keywords:** ER stress and homeostasis, protein folding, Immunology, lipid metabolism, combination therapy, chemotherapy resistance, IRE1α, PERK, ATF6, HSPA5

## Abstract

The unfolded protein response is an intricate network of pathways that mediates intracellular and intercellular cancer cell fate. Previous works have focused mainly on the intracellular roles of UPR. Through a systematic review of the novel roles of UPR, we have highlighted the non-canonical roles of UPR on tumor lipid metabolism and its impacts on the immune system. Additionally, we have also deeply characterized current approaches to targeting UPR effectively and proposed different potential combinatorial approaches to targeting this multifaceted pathway. By gaining a more holistic understanding of non-canonical UPR roles and clearly identifying gaps in the understandings of the dual nature of UPR in pro-survival or pro-apoptosis in cancer cells, we can map out different approaches to target UPR more effectively.

## 1. Background and Introduction

The endoplasmic reticulum (ER) is a highly conserved eukaryotic organelle in the cytoplasm. It is responsible for lipid and protein biosynthesis and homeostasis. As part of its role in protein homeostasis, the ER houses numerous chaperone and quality control proteins to fold and process proteins required by the cell [1,2]. The ER has multiple protection layers against ER stress, with the key objective of maintaining protein homeostasis.

The unfolded protein response (UPR) pathway plays a critical role in handling the increased load of proteins entering the ER, maintaining protein quality and homeostasis via both transcriptional and post-transcriptional mechanisms to avoid ER-stress-induced apoptosis [3]. UPR has three main initiators and ER-stress-sensing proteins which individually and interconnectedly activate different signaling pathways: inositol-requiring enzyme 1 (IRE1), Activating Transcription Factor 6 (ATF6), and protein kinase R (PKR)-like ER kinase (PERK). The UPR pathway also involves other ER chaperone proteins, including the heat shock protein family A member 5 (HSPA5, or BiP, or GRP78). These pathways are briefly described in Figure 1. In response to therapeutic chemotherapy and radiotherapy, cancers exploit the UPR differently to promote resistance via the accumulation of pro-survival signals [4,5]. The UPR has also been linked to lipid metabolism and the immune system, with multiple levels of crosstalk between these processes [6].

## 2. UPR

The UPR pathway is activated when cells undergo ER stress and is triggered when there is an accumulation of unfolded or misfolded proteins detected by HSPA5 and IRE1 [7]. Under physiological conditions, HSPA5 binds to IRE1 and PERK individually while HSPA5 unbinds from both IRE1 and PERK after the detection of misfolded proteins to initiate the UPR pathway [8]. HSPA5 also activates ATF6 by unbinding from the Golgi-localization sequence [9].

### 2.1. IRE1—Canonical Signaling

IRE1 is a highly conserved ER protein across yeast, plants, and animals [10]. IRE1 has two isoforms, IRE1α and IRE1β, with different functions [11]. IRE1α is essential for cell survival and embryonic development as IRE1α knockout mice are embryonically lethal, whereas IRE1β knockout mice are still viable [11]. The extent of overlap, if any at all, between the roles of IRE1α and IRE1β has not been fully explored. There have been reports of IRE1β being the direct negative regulator of IRE1α in multiple epithelial cell lines. Under physiological conditions, ER DnaJ-like homolog 4 (ERdj4, an HSPA5 co-factor) suppresses IRE1α activity, by recruiting HSPA5 to form a complex on the N-terminal luminal domain of IRE1α, and monomerizes IRE1α [12]. The HSPA5-IRE1α complex represses IRE1α and the UPR pathway, though HSPA5 itself has a low affinity for IRE1α. As unfolded and misfolded proteins start to accumulate, they compete to bind with IRE1α and HSPA5 disassociates, thus initiating the UPR [13]. Following the disassociation of HSPA5, IRE1α dimerizes and activates the ribonuclease activity on the C-terminal [14] while triggering autophosphorylation at the cytoplasmic kinase domain [15]. The entire IRE1α pathway is described in detail in Figure 2.

Once IRE1α is activated, IRE1α causes the unconventional splicing of the 26-nucleotide intron of X-box binding protein 1 (XBP1), turning XBP1 into a transcription factor that regulates multiple pathways within the ER [16,17,18]. The IRE1α-XBP1 pathway also transcriptionally regulates ER-associated degradation (ERAD), a critical process in ensuring proper protein folding and quality control by degrading mutated or misfolded protein during both physiological conditions and ER stress conditions [19]. For ERAD to initiate, different ER chaperones, such as HSPA5 and ERdj, bind to misfolded proteins to allow IRE1α activation and oligomerization. Various cancers, including prostate cancer, have taken advantage of the IRE1α-XBP1 pathway to promote pro-survival signals. The IRE1α-XBP1 pathway activates the c-Myc signaling pathway, and androgen receptors also directly upregulate the IRE1α-XBP1 pathway to promote castration-resistant prostate cancer’s survival [20]. To inactivate the IRE1α-XBP1 pathway, the unspliced XBP1 forms a complex with IRE1α-XBP1 and is degraded by the proteasome [21]. Different ERAD adaptors (like Suppressor/Enhancer of Lin-12-like or SEL1L) translocate the misfolded proteins for degradation via E3 ligases (such as HMG-CoA reductase degradation protein 1 or HRD1) [22]. After IRE1α oligomerization and the splicing of XBP1, XBP1 transcriptionally activates different ERAD genes, including SEL1L and HRD1 [23,24]. SEL1L can receive misfolded proteins after activation from different sources, including ERAD-enhancing alpha-mannosidase-like protein 1 (EDEM1) [25], then send them to HRD1 for polyubiquitination and, ultimately, proteasomal degradation. IRE1α-XBP1 and SEL1L-HRD1 cross-regulate each other, with SEL1L-HRD1 as IRE1α’s negative regulator [26]. IRE1α is a substrate of ERAD in vitro and in vivo, as HSPA5’s disassociation from the IRE1α complex initiates the degradation of IRE1α via the SEL1L-HRD1 ERAD pathway [27]. The regulatory mechanism behind the fine balance between IRE1α activation versus degradation is yet to be fully elucidated but remains an intriguing cross-regulating feedback loop. The mechanistic picture of how cancers exploit the IRE1α-XBP1-SEL1L-HRD1 ERAD pathway is also yet to be fully deciphered. Different anti-tumor strategies have emerged that target or exploit ERAD [28], including combining salirasib (a RAS inhibitor) with eeyarestatin (an ERAD inhibitor) due to the synthetic lethality between RAS and several ERAD genes (SEL1L, Ube2g2, Faf2, Syvn1, Sel1l, Aup1, and Derl2) [29] and exploiting ERAD to degrade PD-L1 expression on the cancer cell surface [30]. IRE1α also causes protein degradation via the Regulated IRE1-Dependent Decay (RIDD) pathway, independent of the XBP1 pathway [31]. The selectivity in mRNA properties in RIDD-mediated mRNA degradation is not fully understood. Some common themes of RIDD targets include having XBP1-like stem loop structures (though not all mRNAs with stem loop structures are RIDD targets) and the mRNAs target’s translational status being reduced [32]. RIDD targets are also very sequence dependent, with most targets having the consensus sequence of 5′-CUGCAG-3′. The common functional elements of these mRNA RIDD targets are still unclear. Using different amino acid point mutations, Han et al. identified that IRE1α has two modes of activation: a pseudokinase activation for XBP1 splicing and a phosphotransferase activity for RIDD pathway initiation [33]. Cellular homeostasis is dependent on the balance between IRE1α-XBP1 splicing and the activation of different proteins versus the IRE1α-RIDD degradation pathway. Additionally, the choice between pro-survival signaling via the IRE1-XBP1 pathway vs. pro-apoptotic signaling via the IRE1-RIDD pathway is both context dependent and dependent upon the extent of ER stress. The current model, though requiring more extensive validation in vivo, posits that cells trigger the pro-survival IRE1α-XBP1 pathway with UPR activation to clear misfolded proteins under mild ER stress conditions while cells trigger the IRE1α-RIDD pathway under severe, unresolvable ER stress [33,34,35]. The IRE1α-RIDD pathway has also been shown to cleave and reduce the general level of different pro-survival micro RNA (miRNA), most notably miRNA-17, miRNA-34a, miRNA-96, and miRNA-125b. These miRNAs have been shown to repress caspase-2 mRNA translation (a pro-apoptosis signaling); cleaving these miRNAs can lead to the reactivation of caspase-2 translation, leading to cell apoptosis [18,36]. Furthermore, the destabilization of miRNA17 reactivates thioredoxin-interacting protein (TXNIP) [37], activating the NOD-, LRR-, and pyrin domain-containing protein 3 (NLRP3) inflammasome, and leading to caspase-1- and interleukin-1β-dependent apoptosis [38]. Under severe ER stress, IRE1α binds to TNF receptor associated factor 2 (TRAF2) [39]. This complex binds to apoptosis signal-regulating kinase (ASK1), which further activates c-Jun amino-terminal kinase (JNK) and cell apoptosis [40]. Further understanding of this selectivity between pro-survival and pro-apoptosis can allow for the development of therapeutics controlling cell fate in different diseases, including those triggering cell death in therapy-resistant cancers that adapted to the ER stress caused by the radiation therapy or chemotherapy.

In addition to responding to misfolded proteins, IRE1α has been recently shown to detect lipid bilayer stress. By having an amphipathic helix, IRE1α is sensitive to lipid bilayer thickness and packing density [41]. IRE1α also suppresses membrane permeabilization by preventing the oligomerization of Bax and Bak, protecting the cell from cell death [42]. Though not clearly established, it seems likely that different cancers can exploit this membrane-protective characteristic of IRE1α to become resistant to membrane damage caused by chemotherapy, immunotherapies, and high-dose radiation therapies. This shows that IRE1α can initiate UPR via different mechanisms of detecting aberrancies in misfolded proteins and cell membranes.

### 2.2. ATF6

ATF6 has two isoforms: ATF6α, a potent transcription factor, and ATF6β, a less potent transcription factor and potentially an intrinsic inhibitor of ATFα [43]. Under physiological, inactivated conditions, ATF6 exists as oligomers or dimers that are oxidized at the cysteine residue in the luminal domain [44]. This is noticeably different from IRE1α or PERK’s monomer inactivated state [44]. The C-terminal of ATF6 resides in the ER while the N-terminal is in the cytoplasm [45]. Like IRE1α, ATF6 requires the disassociation of HSPA5 to activate its UPR signaling cascade, as overviewed by Figure 3. Dissociation of HSPA5 from ATF6 leads to a conformational change from an oligomeric to a monomeric form [8,9,44] and to the subsequent activation of ATF6 through reduction by Protein Disulfide Isomerase 5 (PDI) [44]. This reduction results in the exposure of Golgi-localization signals (GLS1 and GLS2) on ATF6. Once mobilized to the Golgi, the transmembrane domain of ATF6 is cleaved by site-1 and site-2 proteases (S1P and S2P) [46,47], allowing the cytosolic fragment of ATF6 to translocate to the nucleus. The degree of reduction in ATF6 correlates with its activation level; the more ATF6 is reduced, the better substrate it becomes for S1P, leading to more efficient cleavage and nuclear translocation [48]. Some studies suggest that ATF6 becomes a suboptimal but acceptable substrate for S2P cleavage only after S1P removes approximately 250 amino acids from its luminal domain [44,49]. This sequential cleavage and activation mechanism is unique to ATF6 within the unfolded protein response (UPR) pathway. This amino acid cleavage for nucleus translocation and protein activation is unique to ATF6 within the UPR pathway.

Whether ATF6 is pro-survival or pro-apoptosis in any identified cancer is a topic of much debate as it is very context dependent, even within the same cancer-like colorectal cancers. Coleman et al. reported that activated ATF6 is associated with a poor innate tumor immune response which promotes tumorigenesis and also results in poor patient outcomes [50]. Hanaoka et al. showed ATF6’s role as a biomarker for precancerous dysplasia in colons [51] while Spaan et al. reported that the activation of ATF6 reduces cellular stemness and proliferation in colorectal cancer cells [52]. In cervical cancer, ATF6 is shown to promote cell survival and migration by upregulating the MAPK pathway [53]. Additionally, ATF6’s interactions with its chaperone proteins are essential for cancer cell survival [54]. ATF6 binds with transcription factors such as Yin Yang 1 (YY1) to further activate ER-stress-dependent ER chaperones such as HSPA5, Glucose-Regulated Protein 94 (GRP94), and ERdj3 [54]. These heat shock proteins and ER chaperones promote various pro-survival pathways, including the PI3K/AKT pathway, and will be further discussed below [55]. On the other hand, ATF6 activation also stimulates other pathways that can lead to cell apoptosis. In preadipocytes, miR103-107 targets WNT family member 3a, leading to ATF6 binding to B-cell lymphoma 2 (bcl2) which, in turn, initiates apoptosis signaling [56]. The overexpression of ATF6 leads to the upregulation of pro-apoptosis WW binding domain protein 1 (WBP1) and the downregulation of pro-survival myeloid cell leukemia sequence 1 (mcl1) [57]. Lastly, ATF6 induces CHOP through crosstalk with the PERK/ATF4 pathway [54,58], and a molecular dynamic simulation suggests the activation of CHOP-mediated apoptosis [59]. With this level of promiscuity in downstream responses, not surprisingly, the selectivity between proliferation and cell death activation by ATF6 remains a mystery, especially in the case of colorectal cancer.

### 2.3. PERK

Protein Kinase R-like Endoplasmic Reticulum Kinase (PERK) is a protein kinase that is also activated by disassociating from HSPA5 upon ER stress [60]. PERK undergoes auto-phosphorylation after activation as it binds and phosphorylates other proteins in different downstream signaling pathways. Like IRE1α, PERK oligomerizes after its activation and binds to different partners [60,61]. This is elucidated clearly in Figure 4.

PERK directly binds to nuclear factor erythroid 2-related factor 2 (NRF2) after activation and phosphorylates NRF2. While the exact site of NRF2 phosphorylation remains unclear, the phosphorylation of NRF2 leads to its nuclear translocation where NRF2 promotes other downstream signaling pathways as a transcriptional activator [62,63]. Under physiological conditions, NRF2 is bound and negatively regulated by Kelch-like ECH-associated protein (Keap1). Keap1 is an adaptor protein to Cullin 3 (Cul3) [63,64], an E3 ligase, that can help the proteasomal degradation of NRF2. Under ER stress and PERK activation of NRF2, Keap1’s cysteine residues undergo post-translational modifications, altering its structural conformations and decreasing affinity for NRF2. NRF2 binds the antioxidant response element (ARE) to transcriptionally regulate different antioxidant enzymes [65]. Cancers often overexpress NRF2 to protect themselves from reactive oxygen species (ROS)-induced apoptosis or ferroptosis. The PERK-NRF2 activation leads to suppression of pro-apoptotic CHOP [65] activation under ER stress. In acute myeloid leukemia, the PERK-NRF2 pathway activates heme oxygenase-1 (HO-1), an anti-inflammatory heme degradation enzyme, and suppresses the phosphorylation of p38. This further decreases the generation of ROS [66], protecting cells from ROS-induced ferroptosis and apoptosis. NRF2 has been recently linked to drug and radiation resistance, particularly in hypoxic tumors. Under hypoxic conditions, NRF2 binds to hypoxia-inducible factor 1α (HIF1α) [67,68], a known regulator of metabolic reprogramming and cell survival, and knocking down NRF2 depletes the protein levels of HIF1α. Through these mechanistic understandings of PERK-NRF2’s role in protecting cancer cells from oxidative stress, inhibition of PERK-NRF2 pathway is revealed as potentially an effective strategy to combine with ROS-inducing or ferroptosis-inducing agents for cancer therapies.

PERK also regulates protein translation by phosphorylating the eukaryotic initiation factor 2α (eIF2α) [61]. eIF2 physiologically brings the initiator methionyl-transfer RNA (Met-tRNAi) to the 43S pre-initiation complex [69] to start protein translation in the ribosomes in a GTP-dependent manner. When eIF2 activity is inhibited under ER stress, PERK phosphorylates eIF2α at serine 51 and inhibits its exchange factor (eIF2B) to cause global translation repression, reducing the protein load and alleviating stress. Additionally, PERK-mediated eIF2α phosphorylation also triggers the transcriptional upregulation of ATF4 [70] which has many functions. ATF4 lowers the levels of caspase-3 (a programmed cell death marker) and poly ADP-ribose polymerase (PARP) under hypoxic conditions, contributing in development of therapy resistance within hypoxic tumors [70]. ATF4 can also bind to Fos-related antigen 1 (Fra-1), a well-studied epithelial-to-mesenchymal transition (EMT) transcription factor, to induce CAMP responsive element binding protein 3 like 1 (CREBL1) in a PERK-dependent manner. A key function of ATF4 is as a transcription factor that translocates to the nucleus and induces the transcription of many autophagy-related genes (Atgs). This leads to the formation of the Atg5-Atg12-Atg16 complex, which converts microtubule-associated protein 1A/1B-light chain 3 form 1 (MAP1 LC3-1, unbound form) to MAP1 LC3-2 (membrane bound form), a key step in autophagosome formation and autophagy induction [71]. By engulfing misfolded proteins and delivering them to lysosomes for degradation, autophagosomes protect cells from stress-induced death and maintain cellular homeostasis. As such, the phosphorylation of eIF2α is seen as a central node in the integrated stress response that determines cell fate; under mild and transient stress conditions, the induction of autophagy preserves cellular viability and integrity but under severe or sustained stress conditions, ATF4 can also bind to CHOP/GADD 153 and mediate cell death [70]. In human osteosarcoma, PERK-eIF2α relies on Protein Kinase R (PKR) [72] to activate autophagy puncta, another key step in cellular autophagy. PERK-pEIF2α can also protect cells from ROS by increasing glutathione biosynthesis and by preferentially translating ATF4 under hypoxic conditions [73], leading to cell autophagy and survival.

PERK phosphorylates different FOXO transcription factors, including FOXO1 and FOXO3. This leads to enhanced FOXO activity, greater nuclear localization, and insulin resistance in ER stress conditions as well as cell survival and treatment resistance in cancer cells [74,75]. The insulin resistance is mediated by circumventing insulin-mediated Akt activation that usually phosphorylates FOXO and sequesters it in the cytoplasm; PERK phosphorylation of FOXO, however, relocates FOXO to the nucleus and promotes insulin resistance. PERK and FOXO can cross-regulate each other. FOXO3 binds to the PERK promoter and drives transcription. While PERK responds to ER stress, another evolutionarily conserved sensor, general control nonderepressible 2 (GCN2), regulates the cellular response to amino acid deprivation, also acting via FOXO [74,75]. These two sensors can compensate for each other, providing a level of redundancy in the integrated stress response. FOXO also activates bcl2/adenovirus E1B 19 kDa interacting protein 3 (BNIP3), which leads to cell death via mitophagy, wherein damaged mitochondria undergo autophagy through BNIP3’s function as an autophagy receptor [76]. A paradoxical suppression of the PERK-FOXO pathway has been observed with the saturated fatty acid palmitate that is known to activate ER stress and insulin resistance; this is likely due to the activation of CHOP that overwhelms pro-FOXO activity [77]. PERK-FOXO3 phosphorylation also upregulates bcl-2 interacting mediator of cell death (BIM) while synergizing with the downstream CHOP signal to lead to cell death [78].

Lastly, PERK is also a lipid kinase. Diacyglycerol (DAG) is a signaling messenger in cells and a direct substrate of PERK [79]. By regulating DAG, PERK controls the precursor of phosphatidic acid (PA) [80], an upstream activator of Ras signaling with mitogenic properties. Additionally, PA binds to AKT which activates mTOR and other proliferation and migratory signals; this PERK-DAG-PA-AKT pathway is regulated by phosphatidylinositol 3-kinase (PI3K) [81]. Under ER stress, PERK is responsible for the accumulation of PA on the mitochondrial membrane by phosphorylating DAG (further creating PA) and eIF2α (inhibiting translation synthesis of the intramitochondrial PA transporter PRELID1). This PA accumulation on the mitochondrial membrane allows mitochondrial elongation and prevents premature fission [82], leading to cell survival under stress. The PERK-AKT pathway has also been shown to be important in protecting prostate cancer from cell death and inhibiting this pathway potentiates CHOP-mediated apoptosis [83].

### 2.4. HSPA5

HSPA5 (also known as GRP78 or BiP) is an activator of the UPR, via the detection of an accumulation of misfolded proteins [84]. As a peripheral protein, HSPA5 can chaperone numerous proteins while helping maintain protein homeostasis and protein folding quality [85], as detailed in Figure 5. HSPA5 has a nucleotide-binding domain and a substrate binding as well as an ER translocation N-terminal and an ER-retrieval KDEL region C-terminal [7]. HSPA5 canonically restricts proteins from aggregating in an ATP-dependent manner, regulated by its own co-factors and chaperones J protein and nucleotide exchange factors [86,87].

HSPA5 has been shown to chaperone proteins to different locations in response to ER stress. HSPA5 primarily functions to transport newly synthesized and misfolded proteins within the ER lumen to help them fold correctly [88]. In non-small-cell lung cancer cells with overexpressed or mutated EGFR, HSPA5 is readily located within the nucleus, where it transcriptionally activates the EGFR promoter by binding to the transcriptional suppressor ID2 [89]. Under ER stress conditions, HSPA5 unbinds from IRE1α, PERK, and ATF6 (due to its intrinsic low affinity to these proteins) [90] to activate UPR. The unbound HSPA5 is typically transported back to the ER by the transmembrane KDEL receptor (KDELR) [91] via coat protein complex I (COPI) vesicular transport, regulated by KDELR1 [92,93]. However, under ER stress conditions, the KDELR-mediated retrieval function may be compromised, leading to KDELR proceeding to the secretory pathway and reaching the cell surface. In different cancers, cell surface HSPA5 (csHPSA5) is indicative of ER stress; csHSPA5 then acts as a receptor for various pro-survival pathways [94]. After thapisgargin treatment in human rhabdomyosarcoma cells, csHSPA5 binds to Cripto-1, a crucial epithelial–mesenchymal transition inducer [95], to reduce anti-proliferative TGF-β signaling and SMAD2 phosphorylation [94,96]. In prostate cancer, csHSPA5 binds to α2-macroglobulin (α2M), a known proteinase inhibitor, to phosphorylate and activate PAK-2, a known mediator of angiogenesis, migration, and metastasis [95]. This csHSPA5-dependent activation of PAK-2 leads to cell proliferation and the overall survival of prostate cancer cells [97]. HSPA5’s ability to translocate to different locations in the cells seems to promote cell survival and therapy resistance in different cancers, though further understandings of the regulation of HSPA5 localization is needed to exploit csHSPA5 as a therapeutic target.

The overexpression of HSPA5 has also been shown to induce varying patient prognoses across different cancers [98,99]. In many cancers, patients with HSPA5-overexpressing tumors have significantly worse prognosis than those with low-expressing tumors [98]. HSPA5 is overexpressed in at least 14 types of cancers, including cholangiocarcinoma [99], colon adenocarcinoma, lymphoid neoplasm diffuse large B-cell lymphoma, esophageal carcinoma, glioblastoma, lower-grade glioma, pancreatic adenocarcinoma, prostate adenocarcinoma, rectum adenocarcinoma, skin cutaneous melanoma, stomach adenocarcinoma, thymoma, uterine corpus endometrial carcinoma, and uterine carcinosarcoma.

HSPA5 also mediates other pro-tumor properties of cancer cells. In gastric cancer, the inhibition of HSPA5 via betulinic acid decreases HSPA5-dependent TGF-β secretion, leading to a decrease in the expression of stemness markers, CD44 and OCT4 [100,101,102], and interleukin 6 (IL6) secretion, and polarization of macrophages from the immunosuppressive M2 phenotype to anti-tumor M1 phenotype [103]. In addition to the regulation of cancer cell stemness and tumor microenvironmental immunosuppression, HSPA5 also plays a role in mediating multidrug resistance. Chen et al. demonstrated that miRNA-495-3p inhibits the HSPA5-mediated activation of macroautophagic receptor Sequestosome 1 (SQSTM1)/p62 in gastric cancer [104] and inhibits the HSPA5-mediated ubiquitination of the mitochondrial ubiquitin ligase activator of NFκB 1 (MUL1), thereby suppressing autophagosome formation and mediating multidrug resistance. In cervical cancer, HSPA5 has been shown to regulate the PI3K/AKT and MAPK pathways to promote proliferation and regulate hypoxia induction [105]. These myriad roles of HSPA5 and csHSPA5 in promoting proliferation, stemness, and immunomodulation provide many nodes for therapeutic intervention within the network of HSPA5-regulated processes.

### 2.5. Non-Canonical Transcription Factors in UPR—CREB3 Family

The cAMP-responsive element binding protein 3 (CREB3) is a component of the non-canonical pathway within the UPR, serving as an ER membrane transcription factor with four homologs—CREB3L1, CREB3L2, CREB3L3, and CREB3L4 [106]. CREB3 shares a similar structure and mechanism of activation as ATF6; they are both transcription factors with a basic leucine zipper structure (bZIP) [107].

When cells undergo ER stress, CREB3 proteins translocate from the ER to the Golgi to be cleaved by S1P and S2P [108]. Similarly to ATF6, this cleavage moves CREB3 into the nucleus where it acts as a transcription factor. CREB3 has a dual-regulation relationship with CREB3 recruitment factor (CREBRF), and CREBRF can be both a positive and a negative regulator of CREB3. In mouse embryonic fibroblast (MEF) cells, CREBRF negatively regulates CREB3 and UPR by binding and degrading CREB3 while bringing CREB3 close to nuclear foci to suppress it [109]. In contrast, a recent study uncovered the positive regulation of CREB3 by non-canonical CREBRF in Neuro2A cells [110]. A holistic understanding of the CREBRF-CREB3 regulatory mechanism remains unchartered; aside from it possibly being context dependent, not much is well-characterized. As Neuro2A cells are used both as neurons and as a neuroblastoma model, we hypothesize that the positive regulation of CREBRF-CREB3 is a cancer-specific mechanism to promote the stability and activation of the pro-tumor CREB3 [111]. Separately, CREB3 is also regulated by the protein kinase CK2, where CK2 phosphorylates the serine 46 residue while not impacting the S1P/S2P cleavage and nuclear localization process [112]. CREB3 activation is also dependent on its N-linked glycosylation at its C-terminal region [113], as the glycosylation supports the S1P/S2P cleavage-dependent activation of CREB3. Among the different canonical UPR inducers, brefeldin A, tunicamycin, and thapsigargin, only brefeldin A causes significant cleavage and activation of CREB3 [114]; the effects of tunicamycin on CREB-H activation was initially reported to include CREB3 induction [115]. However, as tunicamycin has been established as a glycosylation inhibitor and the importance of N-glycosylation in CREB3 activation is becoming better understood, tunicamycin has been shown by others to inhibit the transcriptional activity of CREB-H [113,116]. This highlights the possibility that abrogating glycosylation may offer an alternative therapeutic strategy, which interrupts ER stress responses via CREB3 inhibition to inhibit cancer survival.

CREB3 has been linked to different pro-survival mechanisms, shown in Figure 6. In cervical cancer, CREB3 has been shown to transcriptionally activate homocysteine-inducible, ER-stress-inducible (HERP) [117] by activating Herp’s promoter, a known ERAD ubiquitin-like ER membrane protein to degrade proteins [118] and avoid ER stress-induced apoptosis. Additionally, CREB3 activates and binds to consensus DNA elements such as unfolded protein response element (UPRE) to promote the transcription of different mRNAs [119], including ERAD-enhancing α-mannosidase-like proteins (EDEM), to activate the ERAD machinery in a manner similar to XBP1. Different cancers have taken advantage of CREB3 and its homologs to promote their survival and metastasis. In colon cancer, CREB3 has been reported to bind to the MIR20HG promoter to transcriptionally upregulate MIR20HG and promote cell proliferation [120]. Similarly, knockdown of CREB3 increases p-PERK and ATF4 while promoting glioblastoma apoptosis via BAX and caspase-3 [121]. However, some evidence suggests that CREB3 has anti-tumor properties. In hepatocellular carcinoma, CREB3 transcriptionally upregulated RNA-binding motif protein 38 [122] to suppress Akt phosphorylation. Outside of its transcription-factor activity, CREB3 also competitively binds to insulin receptor substrate 1 (IRS1) to also limit Akt phosphorylation and suppress proliferation [122]. CREB3 also decreases the expression of Slug and Snail, limiting EMT [122]. Therefore, depending upon the role played by CREB3 in specific scenarios, it may be more advantageous to intervene via CREB3 antagonists or agonists.

Different CREB3 homologs exert different pro- and anti-tumor roles, elucidated in Figure 7. CREB3L1 has been shown to bind to the promoter region of different angiogenesis and pro-tumor genes, including FGFbp1 and pleiotrophin, to suppress their expression and block tumorigenesis, survival, and angiogenesis [123]. On the other hand, CREB3L2 is implicated in different cancers as a pro-tumor proliferation signal. In glioma, CREB3L2 activates the transcription factor Activating Transcription Factor 5 (ATF5) by binding to its promoter region. This in turn simulates Myeloid Cell Leukemia sequence 1 (MCL1), a pro-survival signal, to promote glioma growth [124]. In triple-negative breast cancer, the cleavage of the C-terminal fragment of CREB3L2 does not affect the sonic hedgehog pathway in cancer cells but activates this pathway in infiltrating T cells [125]. In CD8+ T cells, this sonic hedgehog activation blocks T cell receptors, while in CD4+ T cells the sonic hedgehog activation increases the expression of IL-4, suppresses CD8+ T cells, and polarizes macrophages into M2 immunosuppressive phenotypes [125]. This paves the way for sensitizing patients to immune checkpoint inhibition via inhibiting CREB3L2 in patients whose tumors overexpress CREB3L2-high. CREB3L2 is also mutated in thyroid carcinomas, where it is fused with peroxisome proliferator-activated receptor γ (PPARγ), a master regulator of fatty acid storage in adipocytes and glucose metabolism in cancer cells, to promote thyroid tumor growth [126].

CREB3L3 and CREB3L4 are novel homologs of CREB3 with conflicting pro- and anti-tumor roles. Chin et al. reported an abundance of CREB3L3 in benign liver cells where it binds to CRE and ATF6 enhancer elements to promote transcription while preventing hepatocellular carcinoma (HCC) cells from entering S phase [127]. However, more recently, Cho et al. reported that hepatitis virus B infection in HCC causes ER-stress-dependent CREB3L3 activation to induce cJun and activate AP-1 target genes. This, in turn, causes CREB3L3-dependent HCC cell proliferation, illustrating the duality of CREB3L3 in HCC [128]. More research is needed to elucidate the regulatory mechanisms of CREB3L3, particularly in the context of non-hepatitis-B-induced HCCs. In breast cancer, CREB3L4 has been shown to bind to the promoter region of proliferating cell nuclear antigen (PCNA), with knockdown of CREB3L4 increasing cell apoptosis and cell cycle arrest [129].

### 2.6. UPR in Cancer Biology as a Whole

The UPR pathway can mediate both pro- and anti-survival pathways in different cancers, with select branches promoting cell survival that can be targeted therapeutically. As described above, branches of PERK and HSPA5 largely aid tumorigenesis and hold enormous potential as therapeutic targets. The list of cancers overexpressing HSPA5 is discussed above; Oncomine and TIMER databases indicate that PERK is overexpressed in breast cancer, liver cancer, lung cancer, gastric carcinoma, lymphoma, thyroid cancer, leukemia, and head and neck squamous cell carcinomas [130]. As different pathways regulated by PERK and HSPA5 contribute to both cancer progression and therapy resistance, attempts to target PERK and/or HSPA5 individually, jointly, or with other potential targets can yield higher efficacy, particularly in cancers overexpressing HSPA5 or PERK.

IRE1α and ATF6 pathways have a very context-dependent role in tumor progression that depends on the extent of ER stress involved. Understanding the severity of ER stress in patient clinical settings can be challenging as there is no objective quantification, and, hence, predicting how interrupting a given pathway will modulate cancer cell survival is challenging. Different cancers can overexpress IRE1α to mediate progression; for example, luminal breast cancer overexpression of IRE1α leads to the RIDD-mediated degradation of tumor suppressor miR-3607 and overexpression of oncogene GTPase RAB3B [131], and castration-resistant prostate cancer overexpression of IRE1α activates IL-6 and the androgen receptor positive feedback loop [131]. On the other hand, inducing IRE1α overexpression increases immunosurveillance against colorectal cancers in vivo [132]. The pro-apoptotic role of ATF6 has been extensively discussed above, with its interactions with bcl2 being predominant. The dichotomous relationship pro- and anti-tumor effects of IRE1α and ATF6 warrant scrutiny to ensure that biomarker-driven strategies are advanced judiciously in specific cancer populations such that cancer pro-survival pathways are not triggered accidentally.

## 3. Novel Impacts of UPR Outside of Protein Homeostasis

Outside their canonical roles of ensuring proper protein folding, resolving ER stress, and retaining protein homeostasis, UPR proteins have been shown to influence other aspects of cellular function. We describe some of these roles in the following sections.

### 3.1. UPR and Immune Cells

UPR has been implicated in a number of ways in the genesis and function of immune cells. This is described in Figure 8. During and after the differentiation of naïve T helper cells into Th2 cells, both IRE1α and spliced XBP1 are overexpressed, leading to an upregulation of different signaling cascades that can be broadly divided into the following three functional categories: ER stress resolution, cell proliferation, and cytokine production. IRE1α-XBP1 stimulates interleukin-5 (IL-5) and interleukin-13 (IL-13) [133], mediating a proliferative drive; however, these pro-inflammatory cytokines [134] can also be counterproductive, promoting lung metastasis [135] and glioblastoma invasion [136]. In different cancers, the presence of Th2 cells in the tumor immune microenvironment can induce the polarization of anti-tumor immunostimulatory M1 macrophages to pro-tumor immunosuppressive M2 macrophages [137] via IL-5 and IL-13, and can suppress the infiltration of CD8+ T cells [138], thereby promoting tumor growth. IRE1α has also been shown separately to cause the polarization of M1 to M2, and the upregulation of PD-L1 in different tumors and white [139,140] adipose tissues. IRE1α also upregulates c-myc to promote natural killer cell (NK cell) proliferation [141,142]. This induction is observed in RNA-seq data in both prostate cancer [143] and non-small-cell lung cancer (NSCLC). IRE1α knockout tumors demonstrated an increase in the abundance of CD8+ T cells and NK cells while demonstrating a lower abundance of immunosuppressive regulatory T cells (CTLA4+ and PD-1+ Tregs). There is more evidence of IRE1α suppressing different pro-inflammatory anti-tumor interferon signaling [143] and promoting the production of prostaglandin E2 (PGE2) [144], a known promoter of Treg differentiation [145], in prostate cancer and NSCLC, respectively. These observations open up the possibility of combining anti-IRE1α inhibition with immunotherapy approaches in traditionally immune-cold cancers.

The innate immune system harbors a key defense mechanism against genotoxic damage and/or extrinsic invasion via the cytosolic sense of DNA in the cell’s cytoplasm, cyclic GMP-AMP synthase (cGAS), and its downstream effector, stimulator of interferon genes (STING). When cGAS senses ectopically localize DNA in the cytosol, it activates the STING pathway to trigger the transcription of multiple inflammatory genes to elicit a type I interferon response [146]. Situated downstream of the cGAS-STING pathway, IRE1α-XBP1 expression has been shown to rescue cancer cells in B-cell malignancies from STING agonist-induced apoptosis [146]. XBP1 also mediates immunosuppressive phenotypes intrinsically in cancer cells. XBP1 acts as a transcription factor for 3-hydroxy-3-methylglutaryl-CoA synthase 1 (HMGCS1), HMG-CoA reductase (HMGCR), 7-dehydrocholesterol reductase (DHCR7), and 24-dehydrocholesterol reductase (DHCR24), vital genes in the biosynthesis of cholesterol [147]. In melanoma and colorectal cancer, XBP1 activation leads to an increase in cholesterol production, which is secreted from extracellular vesicles to myeloid-derived suppressor cells (MDSCs). The cholesterol uptake within MDSCs promotes their activation, proliferation, and immunosuppression, to promote tumor growth [147]. The IRE1α-XBP1 pathway also polarizes tumor-associated macrophages (TAMs) via macrophage inducible Ca2+-dependent lectin receptor (Mincle)-dependent ER stress in an immunosuppressive pro-tumorigenic manner. This pro-tumorigenic polarization is further reinforced by changes in lipid composition; IRE1α-XBP1 and IRE1α-STAT3 are both simultaneously activated to ensure TAM survival in favor of the tumor [148]. This proposes a novel yet crucial crosstalk between the UPR, lipid, and tumor immune microenvironment that needs to be explored further. Another example of UPR-mediated lipid–immune cell crosstalk is the role of XBP1 in interfering with the homeostasis of tumor-associated dendritic cells. These dendritic cells can accumulate 4-hydroxynonenal (4-HNE, a lipid peroxidation byproduct) and constitutively activate XBP1 to upregulate different triglyceride biosynthetic genes, including 1-acylglycerol-3-phosphate O-acyltransferase 6 (Agpat6), fatty acid synthase (Fasn), Stearoyl-CoA Desaturase-2 (Scd2), and Lysophosphatidic acid receptor 1 (Lpar1), leading to an accumulation of triacylglycerides (TAGs). TAGs are prone to becoming oxidatively truncated [149]; oxidatively truncated TAGs and other lipids have been shown separately to inhibit the antigen cross-presentation capabilities of dendritic cells [150]. This XBP1-dependent disruption of dendritic cell antigen presentation functions inhibits local T cells’ anti-tumor responses [149].

PERK also plays a role in modulating the tumor immune microenvironment. PERK modulates MDSCs into immunosuppressive phenotypes [151]. PERK deficiency leads to decreased NRF2 signaling (via lower phosphorylation of ATF4) [152] and the increased production of type I interferon via an upregulation of the STING pathway caused by an accumulation of cytosolic mitochondrial DNA [151]. The upregulation of type I interferon and STING has been shown to create potent anti-tumor effects in different contexts [153]. The role of PERK in suppressing type I IFN shows promising potential in combining anti-PERK with immunotherapy. Under acute ER stress induced by thapsigargin, PERK binds to JAK and activates the STAT3 transcription factor, promoting the expression of C/EBPδ in ductal carcinoma cell lines [154]. The PERK-dependent activation of C/EBPδ creates prolonged activation of CXCL8 (also known as IL-8) and CCL20 [154], two well-known tumor-promoting chemokines that have been shown to recruit regulatory T cells and polarize macrophages into immunosuppressive M2 phenotypes [155,156]. PERK-dependent eIF2α phosphorylation is also required for NF-κB activation [157]. PERK is also central to the metabolically driven epigenetic modification that is required for M2 polarization, by upregulating phosphoserine aminotransferase (PSAT1), serine biosynthesis, and lipid oxidation via ATF4 [158]. Taken together, multiple lines of evidence point to an immunosuppressive role of PERK; novel combinations of PERK-directed therapies with conventional therapy and/or immunotherapy warrant exploration in preclinical and, potentially, clinical contexts.

### 3.2. UPR and Lipid Metabolism

Lipid synthesis, storage, and transport are heavily affected/modulated by UPR, which is clearly detailed in Figure 9. Beyond unfolded proteins overwhelming the ER lumen during stress, changes in the balance between cholesterol and phospholipids in the ER membrane due to metabolic disturbances (obesity, insulin resistance, hepatic steatosis, etc.) also activate the UPR via activation of the IRE1α-XBP1 pathway. This increases phospholipid synthesis for membrane biogenesis, and the activation of the PERK pathway for fatty acid and triglyceride synthesis for lipid storage [159]. We outline the key elements of UPR that maintain and/or restore lipid homeostasis, especially under ER stress conditions.

IRE1α is implicated in lipid rewiring to promote tumor progression and therapy resistance in different cancers. In non-small-cell lung cancer, IRE1α drives c-myc activation. c-myc is shown to activate sterol regulatory element binding protein 1 (SREBP1), which then binds to the multidrug resistance protein 1 (MRP1) promoter and increases MRP1 expression, leading to MRP1-mediated drug efflux and chemotherapy resistance [160]. In triple-negative breast cancer, the inhibition of IRE1α RNAse is shown to increase levels of triacylglycerols (TAGs) and polyunsaturated fatty acids, and to lower levels of diacylglycerols (DAG) while upregulating sterol regulatory element-binding transcription factor 1 (SREBF1), Diacylglycerol O-acyltransferase 2 (DGAT2), and Lipase E (LIPE) genes, hinting at a pivotal role of IRE1α in suppressing TAG biosynthesis. IRE1α-mediated RIDD activity degrades DGAT2 mRNA [160,161]. IRE1α also regulates de novo lipogenesis, a vital process for cancer cell membrane production that aids cancer cell proliferation and aggressiveness [162] in c-myc-driven cancers to promote cellular survival. The IRE1α-c-myc axis has been implicated in the upregulation of lipid synthesis genes, including HMGCR1, HMGCS1, ACLY, ACACA, FASN, and SCD1, with the inhibition of IRE1α suppressing stearoyl-CoA desaturase 1 (SCD1) expression the most [163]. Furthermore, XBP1s binds to the proximal SCD1 promoter to activate it while SCD1 is independently transcriptionally regulated by c-myc [163]; the activation of SCD1 converts saturated fatty acids (SFA) into monounsaturated fatty acids (MUFAs) and is crucial in early cell cycle progression by making phospholipid membranes [164]. In pancreatic cancer, the upregulation of Fasn via the IRE1α-c-myc pathway, and the NR1H2 and NR1H3 liver X receptor pathway also drives chemotherapy resistance [165]. MUFA also suppresses ferroptosis [166]. IRE1α-XBP1 regulation of SCD1 hints at a ferroptosis-suppression role via upregulating MUFAs.

PERK is also activated by lipid stress to promote lipogenesis. In the liver, dysregulated lipogenesis seen in non-alcoholic fatty liver disease (NAFLD) leads to steatosis and ultimately the progression to HCC [167,168]. PERK regulates fatty acid synthase [169] (FAS, a key de novo lipogenesis enzyme [170] required to make fatty acids for cancer cell membranes), ATP-citrate lyase (ACL, another important enzyme that creates acetyl-CoA for fatty acid and cholesterol synthesis [171]), SCD1, and SREB1. PERK activates these enzymes by phosphorylating eIF2α, thus effectively suppressing insulin-inducible gene (Insig1) and SREBP cleavage-activating protein (Scap). The Insig1-Scap complex inhibits ability of Scap to chaperone SREBP to the Golgi, where SREBP has to be cleaved by site-1 and site-2 proteases to activate its transcription factor properties for increased fatty acid and cholesterol synthesis [169]. The PERK-dependent phosphorylation of eIF2α is a vital mediator of hepatic steatosis [172]. Deliberate dephosphorylation (and, hence, the attenuation of eIF2α activity) of eIF2α via the overexpression of eIF2α phosphatase (GADD34-overexpression) has been shown independently to suppress the lipogenesis (and steatosis) transcriptional regulator Peroxisome Proliferator-Activated Receptor (PPAR), and C/EBPα and C/EBPβ, with the phosphorylation of eIF2α promoting C/EBP expression [172]. In a separate study of hepatoblastomas, the ER stress inducer thapisgargin induced the overexpression of HSPA5 and the increased expression of SREBP-1c isoform, leading to upregulation of FAS, ACC1, de novo fatty acid synthesis, and steatosis [173]. All of these pro-hepatic steatosis, PERK-dependent pathways indicate a need for therapeutic inhibition of PERK for NAFLD patients to prevent progression to HCC.

However, SREBP, FAS, and SCD1 are also regulated by other ER stress UPR proteins, independent of PERK, to suppress liver steatosis. Under severe ER stress (indicated by an upregulation of CHOP) induced by tunicamycin, IRE1α and ATF6a pathways are activated to slow down hepatic steatosis via the upregulation of CHOP to suppress different transcriptional regulators of lipid homeostasis including C/EBPα (lipogenesis), PPARα (fatty acid oxidation), Pgc1α (fatty acid oxidation regulator), Srebp1 (lipogenesis), and ChREBP (another coregulator of lipogenesis alongside Srebp1 [174]) in mice. In this scenario, IRE1α has to be phosphorylated at the
${Ser}^{724}$ (one of the three kinase regions of IRE1α [175]) to promote liver steatosis. Of note, the CHOP-mediated suppression of steatosis and the tumor progression driven by IRE1α in liver steatosis models contradict its positive upregulation of lipogenesis in triple-negative breast cancer and pro-tumor proliferation/chemotherapy resistance in NSCLC discussed above. Once again, lipid reprogramming by IRE1α seems to play contradictory tumor-promoting and tumor-suppressing roles, necessitating a cautious approach to targeted therapeutics.

Evolving evidence in the literature points to the role of ATF6 activation in the reprogramming of lipid metabolism in cancer. In colorectal cancer, Coleman et al. noted acute (and chronic) ATF6 activation as leading to the upregulation of Fasn and fatty acid biosynthesis, with lysophospholipids and long-chain fatty acids (LCFA) being the most prominent lipid metabolites [176]. Across their panel of fatty acids studied, the very-long-chain fatty acid (VLCFA) species were the most prominent, with fatty acid elongation attributed to ATF6 activation. The accumulation of VLCFA and Fasn upregulation causes microbial dysbiosis and tumor formation in mouse colorectal cancer models [176,177]. The mechanistic insights into how UPR regulates cancer cells’ intracellular lipid metabolism call for more holistic, context-dependent understandings of this intricate relationship, while probing for hints regarding potential combination therapies targeting both UPR and lipid metabolism pathways.

### 3.3. UPR and ER-Phagy

ER-phagy is a recently discovered process in which parts of the ER undergo autophagy mediated by autophagosomes under ER stress. Once thought to be a spontaneous process, ER-phagy is now considered highly regulated due to discoveries regarding ER-phagy receptors as also tightly connected to ER stress and UPR [178]. There is no consensus on whether ER-phagy plays a pro-survival or pro-apoptosis role; recent research has revealed how UPR and ER-phagy have an intertwined relationship [179].

The role of UPR–ER-phagy relationships in tumor survival or apoptosis is not clearly understood. In breast cancers, it has been established that hypoxia induces both UPR and ER-phagy to maintain ER homeostasis, in which HSPA5 binds to Family with sequence similarity 134, member B (FAM134B) to target [180] the damaged areas of the ER into autophagosomes, while the loss of FAM134B decreases the viability of breast cancer cells. On the other hand, the small molecule Z36 has been shown to upregulate FAM134B, Microtubule-associated protein 1 light chain 3 (LC3), Autophagy-related protein 9 (Atg9), and excessive ER-phagy, consequently further triggering ER stress, activating UPR, and leading to cell death [181]. It remains to be determined whether FAM134B will prove to be a potential therapeutic target, as FAM134B could play contrasting roles in different malignancies. Additionally, UPR and ER-phagy seem to be cross-regulating each other without clear indication regarding which is upstream. Liao et al. demonstrated how excessive FAM134B activity increases ER-phagy and ER degradation, leading to further activation of UPR by increasing IRE1α, XBP1, CHOP, and PERK [181]. However, Zhao et al. showed IRE1α upregulates Endoplasmic Reticulum Protein 1 (Epr1), an ER protein that bridges Autophagy-related protein 8 (Atg8) with VAMP-associated proteins (VAP) for DTT-induced ER-phagy [182]. Similarly, XBP1 has been shown to upregulate methyltransferase-like 3 (METTL3) and METTL4 to enrich m6a methylation and thus promote the mRNA stability of Calcium Binding And Coiled-Coil Domain 1 (CALCOCO1) and p62 (known ER-phagy regulators) in breast cancers [181].

## 4. Therapeutically Targeting UPR Pathways in Cancer

The UPR pathway offers many potential nodes for interruption- or intervention-related therapeutic purposes. The preponderance of the literature suggests that the induction of ER stress and activation of a robust UPR can be exploited to trigger apoptosis selectively in cancer cells. If this selectivity is validated, targeting the UPR pathway has therapeutic potential to eliminate tumor cells just as effectively as conventional treatments, such as chemotherapy or radiotherapy, while limiting apoptotic effects only to the cancerous site and reducing treatment toxicity. The current literature, focusing on manipulating the UPR for therapeutic purposes in cancer treatment, has identified many potential anticancer agents, each targeting a different arm involved in UPR signaling, summarized by Table 1 [183]. Prior research, as detailed extensively above, has demonstrated that whether the cell survives or dies with UPR is highly context dependent (i.e., most likely varying by cancer type and/or treatment type) and intensity dependent (i.e., varying by severity and duration of stress). As such, any inhibitors or potential therapeutic agents targeting UPR pathways must be tightly controlled [183]. Outlined below are potential avenues for therapeutic intervention.

### 4.1. Overcoming Chemoresistance Regulated by UPR Using Combinatorial Approaches

One of the most well-documented therapeutic utilities of UPR and ER stress is the opportunity to combat resistance to traditional anti-tumor regimens. Many molecular pathways by which UPR induces chemoresistance have been identified. Chemo-resistant cancer stem cells have been shown to hijack UPR to maintain growth and proliferation despite treatment with chemotherapy. Suppressing core UPR branches, however, sensitizes cancer cells to chemotherapy [5,184], even though UPR regulatory factors may express paradoxical roles in regulating cell survival and/or cell death [185].

The family of NRF transcription factors, particularly NRF2, is a major player in regulating UPR and is highly involved in driving chemoresistance. Interestingly, NRF2 plays a dual role in cancer; NRF2 expression can dampen tumor growth, but prolonged expression of NRF2 can result in pro-tumor responses such as chemoresistance. Therefore, targeting NRF2 to address tumor growth and tumor resistance requires precision and a clear understanding of the mechanism underlying its dual role [185]. For instance, the anticancer agent camptothecin was evaluated in early clinical trials for many cancers, including malignant melanoma and advanced gastric malignancies [186]. By inhibiting the NRF2-ARE activation pathway in a concentration-dependent fashion, camptothecin sensitized HCC cells to chemotherapy and anti-tumor drugs [187,188]. Additional compounds targeting NRF2 have been identified, but most are natural compounds that have yet to be evaluated for clinical significance. Other UPR-targeted pharmacological agents that synergize with anticancer drugs to amplify their effects include integrated stress response inhibitor B (ISRIB) [189], which reverses eIF2α phosphorylation and enhances gemcitabine-induced pancreatic cancer cell death, MKC-3946 [190], which enhances bortezomib or arsenic trioxide tumor-killing effects in acute myeloid leukemia cells by inhibiting IRE1α ribonuclease (RNase), and epigallocatechin gallate (EGCG), which directly targets HSPA5 to resensitize glioma cells to temozolomide chronotherapy [191]. Data gathered on the effects demonstrates significant promise for future clinical development [192,193]. The NRF2-ARE axis is also known to drive radio resistance in various cancers. In colorectal cancer, irradiation via KRAS activation led to NRF2 upregulation, which further enhanced 53-BP1 mediated non-homologous end-joining repair in malignant cells [194]. Also, in esophageal squamous cell carcinoma, fractionated radiation is known to trigger ΔNP63α activation, leading to increased NRF2 translocation to the nucleus, and thus driving resistance to therapy [195]. Inhibitors of the NRF2-ARE pathway can therefore be employed as a complement to radiation-induced cell death.

### 4.2. Overcoming Immunotherapy Resistance by Targeting UPR Pathways

Combining pharmacological agents targeting UPR and immunotherapy has also shown promise as a synergistic anticancer intervention [191,192]. For instance, PI3K/AKT inhibitors show clinical significance in treating breast cancer due to common mutations in the PIK3CA and PTEN genes. However, PTEN-deficient breast cancers are resistant to these inhibitors. Combining inhibitors of the histone demethylase KDM4B, such as methylstat and PI3K/AKT inhibitors, upregulates the UPR downstream target ATF4, resulting in UPR activation and the apoptosis of PTEN-deficient breast cancer cells [196]. IRE1α activation impairs mitochondrial respiration and the fitness of intratumoral T cells, thus hindering their tumor-fighting capacity [196]. Prolonged PERK activation hinders CD8+ T cells in ovarian cancer, resulting in less efficient immune targeting [197]. From the foregoing discussion of different branches of the UPR pathway, it seems reasonable to posit that combining immunotherapy with inhibitors of IRE1α or PERK will likely improve therapeutic efficacy. In fact, IRE1α inhibition sensitizes PD-L1-TP53-mutant, immunologically ‘cold’ TNBCs to chemo-immunotherapy, and IRE1α inhibition works synergistically with an anti-angiogenic, anti-VEGF-A antibody [198] to cause tumor regression in TNBC cells [199].

### 4.3. Specifically Targeting the IRE1α Arm

IRE1α signaling is a core branch of the UPR pathway. To highlight the importance of targeting this pathway, the IRE1α-XBP1 pathway has been shown to directly activate c-myc, a key oncogenic pathway in prostate cancer [20]. IRE1α signaling also alters the tumor microenvironment and promotes prostate cancer proliferation [143]. Potential pharmacological targets that interrupt the IRE1α pathway include kinase II inhibitors, RNase inhibitors, and RNase activators [200]. ORIN1001 is a notable IRE1α-XBP1 blocker that is currently progressing in phase II clinical trials for advanced solid tumors (recruiting for advanced breast cancer patients). ORIN1001 is taken orally and targets the RNase of IRE1α, inhibiting the IRE1α-XBP1 pathway and thus potentially inhibiting UPR [201]. In breast cancer, the hormonal therapy drug fulvestrant indirectly inhibits the IRE1α-XBP1 axis, resulting in selective apoptosis [202]. In prostate cancer, fenofibrate activates IRE1α and PERK by acting as a PPAR-γ antagonist [203]. In hematological malignancies, type I and II kinase inhibitors, as well as RNase inhibitors of IRE1α, are deemed promising candidates [204]. Of note, the FDA-approved anticancer drug sunitinib inhibits IRE1α autophosphorylation, diminishing XBP1 splicing in blood cancer cells treated with an ER stress inducer [15,204]. Collectively, these examples provide evidence of the druggability of the IRE1α-XBP1 axis, but the dosage and agent will likely depend on the cancer subtype and patient profile [205]. IRE1α amplification and overexpression is prevalent in cancers such as aggressive luminal breast cancers and glioblastoma [131,206]. Interestingly, up to 65% of patients who responded poorly to ER-stress-directed chemotherapeutics demonstrated an overexpression of IRE1α [206]. Targeting IRE1α, therefore, is a promising strategy. Since the UPR is a ubiquitous cellular response to stress, tumor-selective targeting of the IRE1α axis is unattainable unless inhibitors are selectively delivered/targeted against novel tumor antigens. However, the amplification and overexpression of IRE1α in cancers offers a non-conventional chemotherapeutic target. Therefore, targeting IRE1α in such patient populations could potentially lead to therapeutic benefits with diminished systemic toxicity compared to conventional anticancer agents (taxanes, platinum compounds, and anthracyclines).

### 4.4. Specifically Targeting the PERK Arm

PERK signaling in UPR is involved in many cancers, including colorectal cancer and squamous cell carcinoma, as established above. PERK inhibitors have shown some promise in addressing cancer metastatic burden. The clinical-grade PERK inhibitor (HC4) eradicated quiescent cancer cell micro-metastases in the bone marrow and lungs of HER2+ breast cancers in syngeneic and patient-derived xenograft models [207]. The chemotherapeutic agent doxorubicin blocks cancer proliferation via CREB3L1, which is a crucial downstream mediator in PERK-driven metastasis [208,209]. Only three PERK-specific inhibitors (GSK2656157, GSK2606414, and AMG44) [210,211], to the best of our knowledge, have been selected for clinical development. One of these, GSK2606414 was associated with more side effects, such as enhanced vulnerability to viral infections [212], highlighting the need to carefully evaluate optimal dosing strategies and/or prophylactically treat patients to prevent opportunistic infections. Another of these agents, GSK2656157, showed potent preclinical anti-tumor activity in multiple human xenograft models but also caused on-target off-tumor pharmacologic effects on pancreatic function, resulting in weight loss and diabetes [213]. Additionally, there is evidence that PERK is a tumor suppressor in certain contexts, rendering PERK inhibitors counterproductive and highlighting the need for caution in the clinical pursuit of this inhibition strategy [214]. On the other hand, there may be ways to interweave PERK-targeted therapies into the treatment algorithms of cancer patients if the predictors of response are known beforehand. In colorectal cancer patients, not only was there differential expressions of PERK and HSPA5 in tumor and metastatic tissues but also single nucleotide polymorphisms were predicted to contribute to genetic variants that may influence the susceptibility of targeted UPR agents [215].

Transmissible ER stress (TERS) is a phenomenon recently discovered where cells undergoing UPR can trigger a unique UPR mechanism from one (myeloid) cell to another, promoting a cascade of cell proliferation. TERS is mediated by the differential signaling of PERK, promoting cell survival [55]. More research is warranted to evaluate any therapeutic potential in targeting this epiphenomenon.

### 4.5. Specifically Targeting the EIF2α Arm

The phosphorylation of EIF2α is a hallmark of autophagy and cell death mechanisms, which makes this pathway highly promising for therapeutic targeting. Current chemotherapeutic taxanes, such as paclitaxel, cabazitaxel, and docetaxel, target microtubules and activate the PERK-eIF2α axis, leading to cancer cell apoptosis in metastatic breast, ovarian, and lung cancers. Pharmacological inhibition of PERK blocks taxanes from inducing their anti-tumor effects [216]. Paclitaxel induces ER stress by inhibiting selenoprotein S expression [217] and its excipient triggers the UPR [218]. For cancers resistant to paclitaxel, targeting the EIF2α and ATF4 pathways mitigates drug resistance [219]. In melanoma, docetaxel-induced apoptosis is mediated by the upregulation of ER stress-associated proteins, such as PERK/eIF2α [220]. Resistance to cabazitaxel, used to treat metastatic castration-resistant prostate cancer in combination with prednisone [221], is mediated by the UPR signaling pathways [222] IRE1α and PERK [223], although the underlying mechanisms behind the feedback loop remain unclear. Other chemotherapy drugs such as bortezomib and salubrinal induce eIF2α phosphorylation indirectly [224], resulting in anti-tumor effects, further reinforcing the potential for the combination of chemotherapy and eIF2α in synergistic tumor cell kill. The inhibition of eIF2α through various ATP-competitive compounds (GCN2 inhibitor SP600215 [225,226]; BCR-ABL inhibitors [225,226]; imidazolo-oxindole PKR inhibitor C16 [227]; PERK inhibitors, etc.) that suppress the integrated stress response can help to overcome drug resistance, illustrating the benefit of multi-targeted approaches.

ATF4 is intrinsically intertwined with eIF2α, as ATF4 expression is directly downstream from eIF2α signaling. While targeting eIF2α as an upstream target of ATF4 could be a potential therapeutic approach, the direct inhibition of ATF4 has not been reported. Nonetheless, multiple potential indirect ATF4 inhibitors function through mechanisms such as kinase inhibition [228]. Further research should be conducted into whether ATF4 is a druggable therapeutic target.

### 4.6. Specifically Targeting HSPA5

The ER-resident chaperone HSPA5 is responsible for tumorigenesis in many organs via overexpression [229] and translocating to the cell surface in many cancer cells. ER stress conditions associated with anticancer therapy further induce HSPA5 upregulation and translocation to the cell surface. This upregulation of cell-surface HSPA5 has cytoprotective effects for cancer cells, making the HSPA5 pathway a promising candidate to target.

Various classes of HSPA5 inhibitors have been identified. HSPA5 antibodies exploit HSPA5 translocation to the cell surface. Clinically, PAT-SM6 is the only HSPA5 inhibitor in phase I clinical trials for tumor therapy, but this trial was terminated due to limited clinical benefits [229]. There are also many small molecule HSPA5 inhibitors identified, both synthetic and natural. Natural compounds acting as HSPA5 inhibitors have low bioavailability and demonstrate limited specificity in targeting HSPA5, making them difficult for clinical applications [229]. Synthetic HSPA5 inhibitors are in development, and their high oral bioavailability makes them relatively suitable for clinic use [222]. Several cancers inherently demonstrate a high expression of cell-surface HSPA5 (cs-HSPA5)—pancreatic, lung, gastric, and esophageal, among others [230,231,232,233]. Also, in glioblastoma, lung cancers, and pancreatic cancer, radiation-induced cs-HSPA5 upregulation has been recorded. In such cases, upon targeting with systemic monoclonal antibody therapies, cs-HSPA5 blockade was shown to enhance the efficacy of radiotherapy in tumor models [230,234]. Simultaneous and sequential cs-HSPA5 targeting, in conjunction with irradiation, therefore presents an enticing one-two punch strategy for managing malignant tumors. Furthermore, such combination therapies would be spatially and temporally selective, thereby minimizing off-target toxicities.

### 4.7. Specifically Targeting ATF6

While ATF6 is a major component of the UPR pathway, the modulation of this pathway remains incompletely explored. The small molecule inhibitor ceapin-A7 has been shown to block ATF6 activation by trapping the full-length molecule in the ER, preventing the generation of the cleaved form that translocates to the nucleus [235,236]. Melatonin and PF−429242 are also ATF6 inhibitors [237]. The experimental use of combined Adriamycin and ATF6 inhibitors sensitizes colon cancer cells that are typically resistant to treatment to cytotoxic effects [238,239]. Another approach uses 4-phenylbutyric acid (4-PBA) as an inhibitor of ER stress by inhibiting the HSPA5-ATF6-CHOP axis and restoring ER function and protein synthesis [240]. Aside from these early efforts to target ATF6, the promise of targeting this arm of the UPR remains largely unfulfilled.

### 4.8. Nanoparticles

Recent developments in nanotechnology have fueled the quest for precision therapeutics targeting key hallmarks of cancer and/or (mal)adaptations of cancers to conventional therapies. Nanoparticles can be used to deliver drugs precisely to address chemoresistance and accelerate cell death [241]. Nanoparticles can exert anticancer effects by ferrying encapsulated drugs selectively to cancers, preferentially targeting cancer cells for the delivery of therapeutic payloads, or transferring surface-bound molecules to cancers for cytotoxicity [242]. In the context of UPR and ER stress, nanoparticles can be utilized to trigger UPR cascades. For instance, graphene oxide-based nanoparticles have been engineered to induce ER stress through the ER stress inducers doxorubicin and cisplatin. These nanoparticles also induced autophagy, suggesting the benefit of targeting the UPR-autophagy crosstalk [243]. Nanoparticles made of different materials, such as gold, silver, iron oxide, manganese, titanium oxide, silica, quartz, and carbon, also demonstrate therapeutic effects possibly mediated through the modulation of the UPR pathway [244]. Notably, solid tumors exhibit preferential drainage and accumulation of nanoparticles (more so in the 100–200 nm size range) in lieu of the EPR (enhanced permeability and retention) effect [245]. Furthermore, surface functionalization strategies have been employed to target tumor tissue. Various nano formulations have also been successfully tested in clinics (liposomal Irinotecan/Onivyde^®^ for pancreatic cancer, liposomal Doxorubicin/Doxil^®^ and albumin-bound Paclitaxel/Abraxane^®^ for multiple solid tumors) [246,247,248]. Likewise, nanoparticle-based delivery approaches can aid in inducing ER stress and apoptosis in tumor tissues preferentially, while minimizing systemic toxicity. Apart from acting as targeted carriers of ER stress inducers, nanoparticles themselves can trigger UPR. This has been demonstrated by Pandey et al. by designing graphene oxide-based doxorubicin and cisplatin NPs that induce ER stress and autophagy, culminating in cell death [243].

**Table 1 cancers-17-03639-t001:** Previous strategies of targeting UPR in cancer therapies.

Agents	Mechanism	Pathway(s) Targeted	References
Camptothecin	Inhibits NRF2 signaling, sensitizing cancer cells (e.g., hepatocellular carcinoma) to chemotherapy and anti-tumor drugs by blocking the NRF2–ARE activation pathway.	NRF2	[186,187,188]
ISRIB	Inhibits eIF2α phosphorylation, enhancing gemcitabine-induced pancreatic cancer cell death.	eIF2α	[189]
MKC-3946	Inhibits IRE1α ribonuclease activity, enhancing tumor-killing effects of bortezomib or arsenic trioxide in acute myeloid leukemia cells.	IRE1α	[190]
EGCG	Targets HSPA5 to resensitize glioma cells to temozolomide treatment.	HSPA5	[191]
KDM4B inhibitors (methylstat)	Inhibit KDM4B, upregulating UPR target ATF4 and triggering apoptosis in PTEN-deficient breast cancer cells.	KDM4B	[196]
PI3K/AKT inhibitors	Used in breast cancer, but combined with KDM4B inhibitors to sensitize PTEN-deficient tumors to apoptosis by inducing UPR and activating ATF4.	KDM4B, ATF4	[196]
IRE1α inhibitors	Target IRE1α signaling to reduce tumor proliferation and alter the tumor microenvironment; effective for both solid and liquid cancers. ORIN1001, a notable IRE1α-XBP1 blocker, is progressing in clinical trials.	IRE1α	[200,201]
Fulvestrant	Indirectly inhibits IRE1α-XBP1 axis and induces selective apoptosis in breast cancer cells	IRE1α	[202]
Fenofibrate	Activates IRE1α and PERK, acting as a PPAR-γ antagonist, in prostate cancer cells.	IRE1α, PERK	[203]
Sunitinib	FDA-approved for blood cancers; inhibits IRE1α autophosphorylation, reducing XBP1 splicing in blood cancer cells.	IRE1α	[15,204]
PERK inhibitors (HC4, GSK2656157, etc.)	Block PERK signaling, preventing metastasis in cancers like HER2+ breast cancer and colorectal cancer; their application must consider side effects such as viral infection susceptibility.	PERK	[207,210,211,212,213]
Doxorubicin	Blocks cancer proliferation via CREB3L1, a downstream mediator of PERK in metastasis. Target of ER stress induction by nanoparticles.	CREB3L1 (PERK)	[208,209]
Taxanes (e.g., paclitaxel and docetaxel)	Activate PERK/eIF2α axis, inducing cancer cell apoptosis by targeting microtubules; resistance can be mitigated by targeting EIF2α/ATF4 pathway.	IRE1α, PERK	[216,217,218,219,220,221,222,223]
GCN2 inhibitors (SP600215, BCR-ABL)	Inhibit ISR (Integrated Stress Response) by blocking EIF2α phosphorylation, which can overcome drug resistance in cancers.	EIF2α	[225,226]
Ceapin-A7	Selectively inhibits ATF6a in UPR, with potential for colorectal and pancreatic cancer treatment.	ATF6a	[235,236]
Melatonin, PF−429242, combined therapy, 4-PBA	Inhibits ATF6, demonstrating potential to sensitize colon cancer cells to cytotoxic effects in combination with Adriamycin; targets HSPA5-ATF6-CHOP axis.	ATF6 and related axes	[237,238,239,240]
HSPA5 inhibitors (PAT-SM6)	Target HSPA5, a chaperone protein involved in tumorigenesis; currently in phase I trials for tumors.	HSPA5	[249]
Nanoparticles (e.g., graphene oxide-based)	Deliver drugs to cancer cells to induce ER stress and autophagy; can encapsulate chemotherapeutic agents (e.g., doxorubicin and cisplatin) for targeted delivery.	N/A	[241,242,243,244,245,246,247,248]

## 5. Limitations and Challenges in Targeting UPR

Targeting individual UPR proteins has been challenging due to their duality in supporting cancer growth versus inducing apoptosis, as well as their localization and crucial role in benign cells. This dichotomy in the UPR, which directs signaling along divergent pathways—proliferation vs. apoptosis—is undoubtedly context, intensity (dose and duration), and tumor type dependent. When fully defined, the UPR pathway direction can be selectively targeted, but clinical efforts to date have not teased out this predilection for a preferred downstream pathway in response to ER stress. Since IRE1α and PERK can respond to therapy-induced stress by triggering pro-tumor proliferative signals or anti-tumor apoptotic signals, there is an unmet need for patient stratification and personalized biomarker development when incorporating UPR inhibitors into clinical workflows and treatment paradigms. For instance, PERK inhibitors have been noted to have high pancreatic toxicities, and earlier PERK inhibitors, such as GSK2606414 and GSK2656157, have been reported to have critical specificity issues [250], as both compounds turned out to be Receptor-Interacting Serine/Threonine-Protein Kinase 1 (RIPK1) inhibitors, affecting cell death pathways [251]. In addition, since the ER stress response is an evolutionarily conserved pathway required for combating other forms of ER stress in normal cells, beyond the adaptive response to cancer therapy, there is a possibility of making patients more prone to contracting infectious diseases.

While several drugs targeting UPR are still in the preclinical testing stage, very few have moved to early-phase clinical trials. In a phase 1 trial (NCT01727778), the monoclonal immunoglobulin M antibody PAT-SM6 targeting the cell surface GRP78/BiP (HSPA5) in subjects with relapsed or refractory multiple myeloma demonstrated good tolerability. However, this trial showed no objective responses, with only 33% of patients achieving stable disease [249]. The IRE1α inhibitor ORIN1001 has been evaluated in patients with advanced solid tumors or in those with relapsed and refractory metastatic breast cancer in a completed phase 1/2, open-label, dose-escalation and dose-expansion study (NCT03950570). The early efficacy results have reported that only 2 of 30 patients achieved partial response (PR) when used as a single agent, and dose-limiting toxicities included thrombocytopenia and rash [252]. Another phase I clinical study (NCT05154201) evaluating ORIN1001 monotherapy and its combination in patients with advanced solid tumors has been completed; however, the results have yet to be published. Similarly, a phase 1a study (NCT04834778) of the PERK inhibitor HC-5404-FU in patients with advanced solid tumors has been completed, and the results of the final safety analysis and clinical responses have yet to be published. Results from these studies in the future will provide insights into the efficacy and safety of UPR-targeting agents.

These findings from preclinical and early clinical studies highlight fundamental challenges in UPR-targeted therapy. The essential role of UPR in normal cells creates a narrow therapeutic window and resistance to therapy, likely due to compensatory mechanisms between the UPR pathways (PERK, IRE1α, and ATF6) and a lack of validated biomarkers for optimal patient selection. Notwithstanding these potential toxicities, with a more holistic understanding of UPR biology and tumor response heterogeneity, selective individual UPR-targeted therapy or combination therapy with other agents like sunitinib, doxorubicin, taxanes, and bortezomib will likely evolve in specific cancers [15,200,210,211,212,213,214,215,216,217].

## 6. Conclusions, Perspectives, and Future Directions

The UPR pathway is a multifaceted evolutionarily conserved response to ER stress that dictates cell fate in different contexts. We have summarized how IRE1α, PERK, and ATF6, the three primary sensors of ER stress, trigger individual signaling cascades after dissociating from the chaperone HSPA5 when misfolded proteins accumulate in the ER. Collectively, these pathways restore protein homeostasis. IRE1α goes through XBP1-mediated ER-associated degradation, TRAF2-ASK1-JNK apoptosis, or RIDD-mediated decay of mRNAs coding proteins accumulating in the ER. ATF6 undergoes cleavage by site-1 and site-2 proteases in the Golgi apparatus, and the active form of ATF6 translocates to the nucleus to transcribe a number of ER chaperone genes and XBP1. Lastly, PERK controls global protein translation through interactions with eIF2 to reduce new protein synthesis and with FOXO to increase the translation of ATF4; in turn, it modulates antioxidant responses, amino acid metabolism, and possibly CHOP-mediated apoptosis. We described recent works that elucidate the non-canonical roles of IRE1α-XBP1 and PERK in mediating cytokine production, macrophage polarization, PD-L1 upregulation, and T-reg- and c-myc-driven NK cell proliferation, demonstrating the potential of combining UPR inhibitors with immunotherapy. We also highlight how IRE1α promotes tumor progression by increasing the production of pro-tumor lipid biosynthesis and de novo lipogenesis. Collectively, the UPR pathway serves to sense the accumulation of unfolded proteins in response to ER stress and to then lead the cell via one of two principal downstream, tightly regulated cell fate choices: repair and restore protein homeostasis, or abort and initiate programmed cell death. The choice seems to be highly context, intensity (duration and severity of ER stress), cancer type, and therapy dependent. Optimizing UPR-targeted therapy will, therefore, need to identify the ideal therapeutic window in each cancer context, accounting for tumor heterogeneity and patient/population diversity. The localization of the UPR signaling cascade is also crucial; the discovery of non-canonical ER-stress-induced cell surface localization of HSPA5 is the first of many steps for more accessible targeting [94,96]. A more nuanced understanding of the (mal)adaptations of cancers to conventional therapies should foster the development of rational, evidence-based, biomarker-driven, personalized combination therapies that exploit vulnerabilities unique to specific cancers in specific contexts. While doing so, it will be important to recognize the unique toxicities of UPR-targeting strategies, namely the high predilection for pancreatic toxicity, due to the dominance of FOXO in the insulin response/resistance, and the potential vulnerability of patients to infections which would have been countered by an intact defensive UPR.

## Figures and Tables

**Figure 1 cancers-17-03639-f001:**
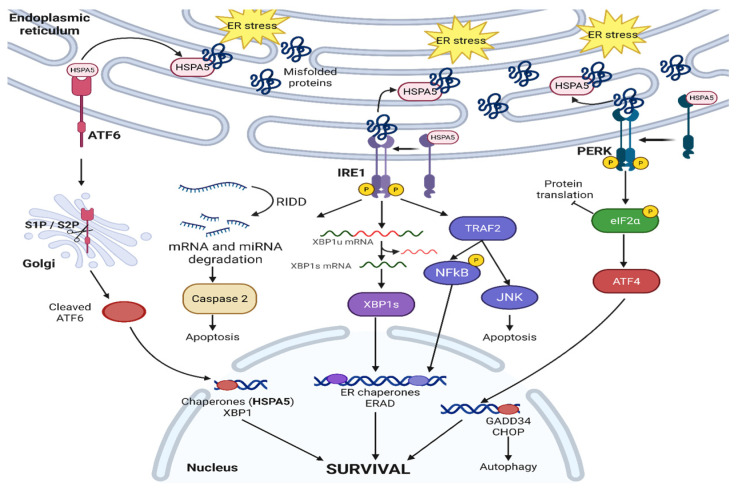
Overview of UPR pathway.

**Figure 2 cancers-17-03639-f002:**
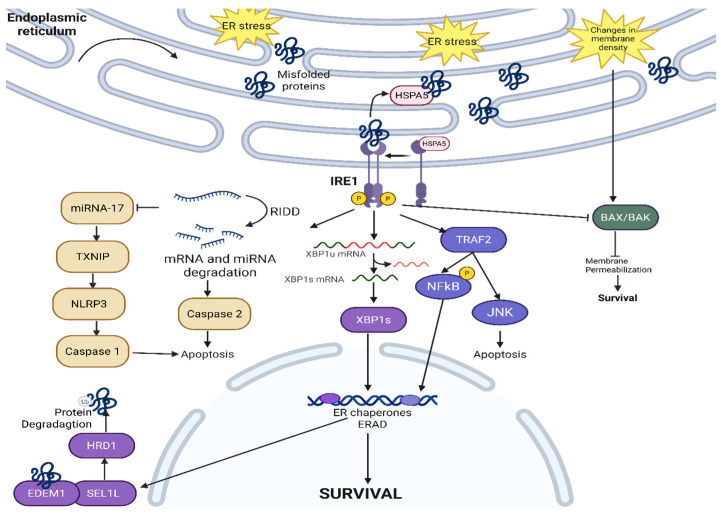
Overview of IRE1 signaling pathway.

**Figure 3 cancers-17-03639-f003:**
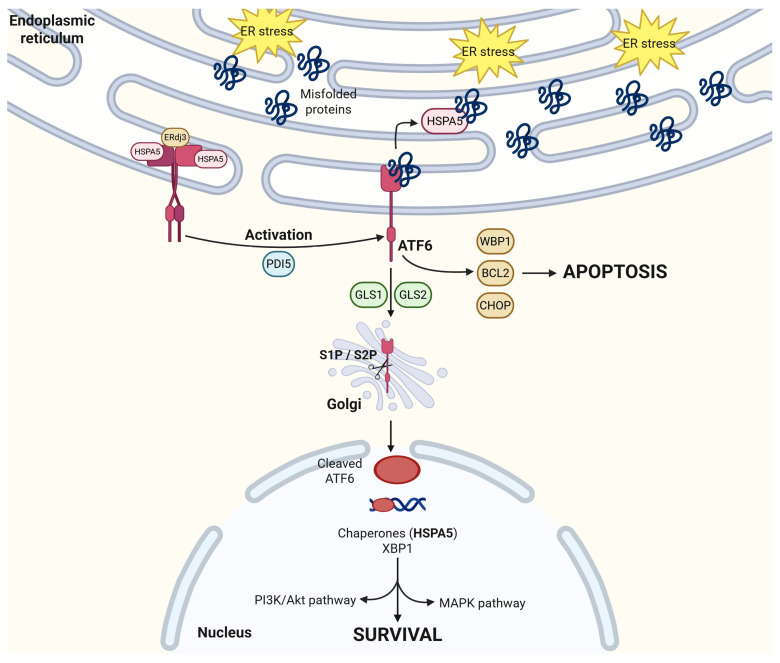
Overview of ATF6 signaling pathway.

**Figure 4 cancers-17-03639-f004:**
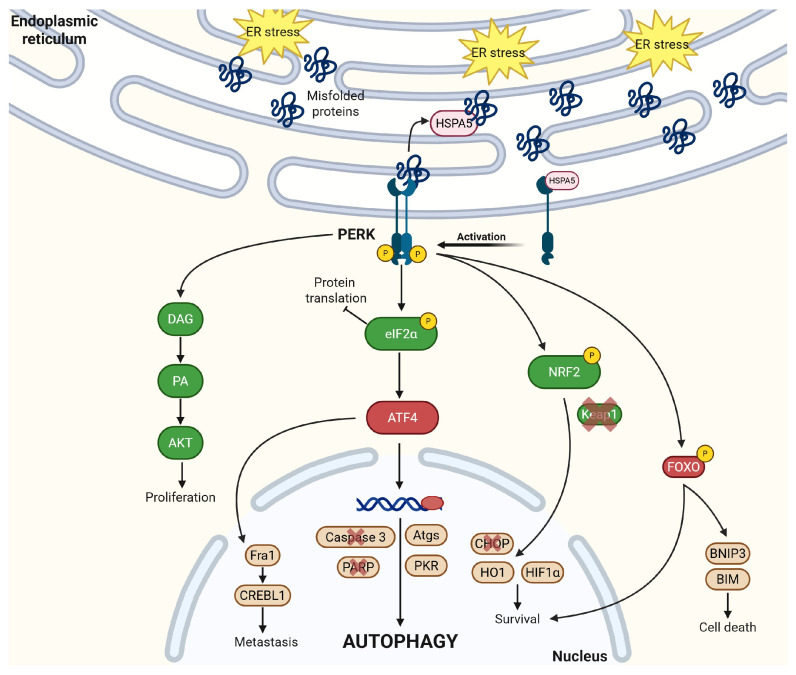
Overview of PERK signaling pathway.

**Figure 5 cancers-17-03639-f005:**
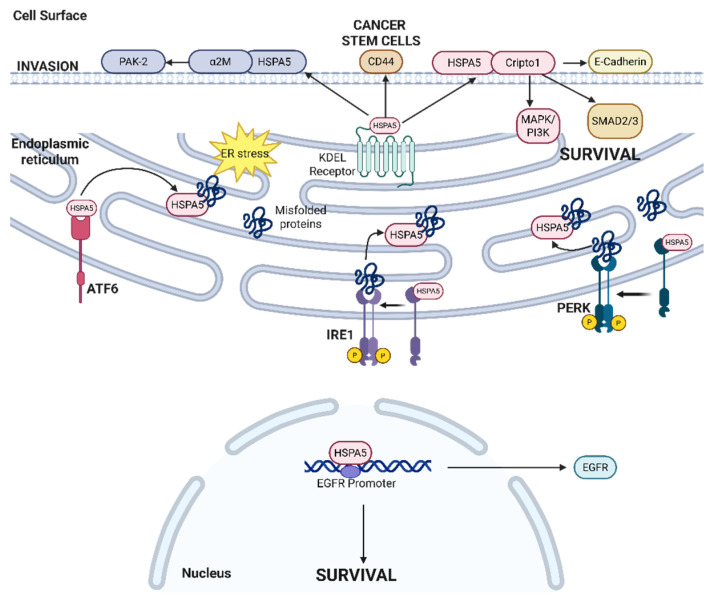
Overview of HSPA5 signaling pathway.

**Figure 6 cancers-17-03639-f006:**
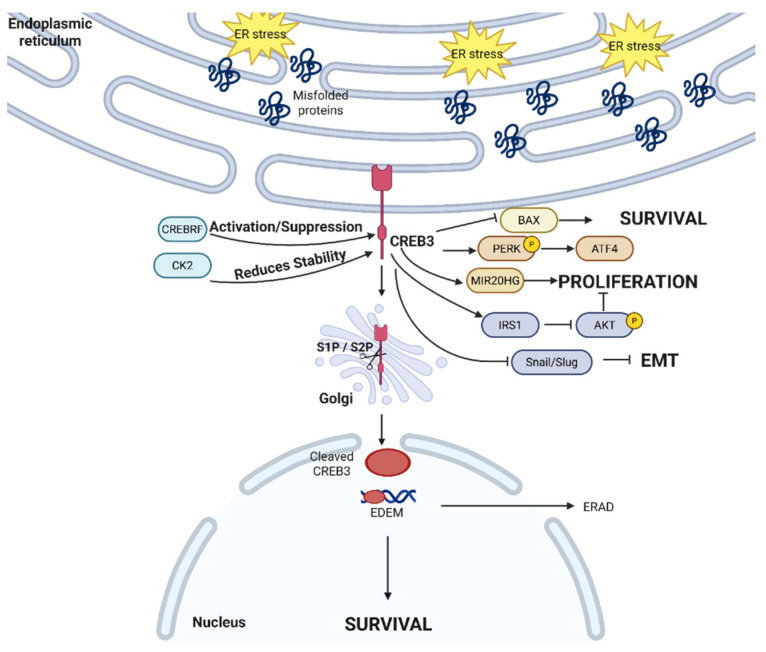
Overview of CREB3 signaling pathway.

**Figure 7 cancers-17-03639-f007:**
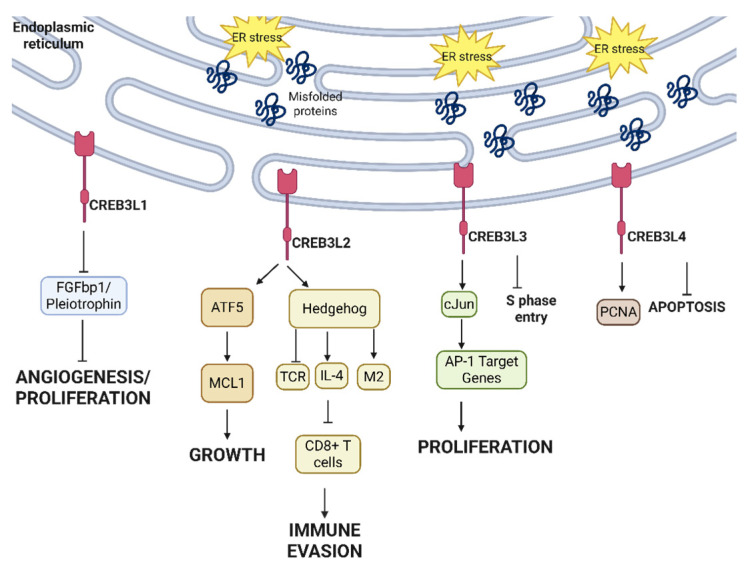
Overview of CREB3L signaling pathway.

**Figure 8 cancers-17-03639-f008:**
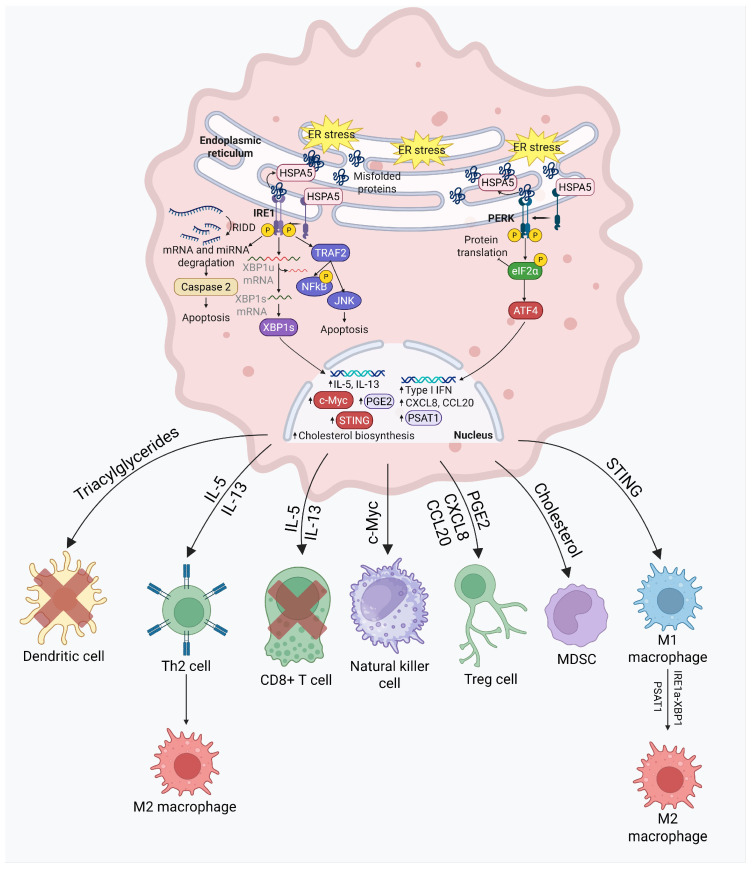
Role of UPR in immune cells.

**Figure 9 cancers-17-03639-f009:**
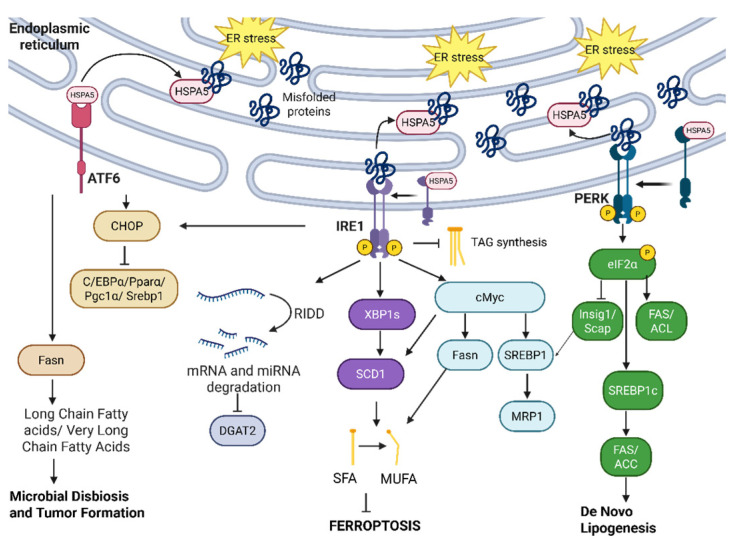
Role of UPR in lipid metabolism.

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
