# Peer review of "The Unfolded Protein Response—Novel Mechanisms, Challenges, and Key Considerations for Therapeutic Intervention"

_cancers, 2025, doi:10.3390/cancers17223639_

Round 1

Reviewer 1 Report

Comments and Suggestions for Authors

The authors have prepared a review that provides an overview of the unfolded protein response, with particular emphasis on its role in cancer cell biology. This is generally a well-written review that covers both basic and clinical aspects and is worthy of consideration for publication in the journal. However, the manuscript requires some improvements.
Addressing the following points may enhance the accuracy and overall quality of the manuscript.

General comments

The numerous figures included in the work are very helpful in understanding the main text. However, what is presented in them does not always correspond well with their descriptions in the text. The authors should carefully compare the content of the figures with the information provided in the text. Also, for clarity, the figure legends should be expanded to include explanations of the most important abbreviations used. In some figures, the labels are too small and unreadable (Fig. 3 and 8)

Attention should be paid to ensure that gene names are always written in italics.

Specific comments

Section: IRE1 – Canonical signaling

For the sake of clarity:

(1) the paragraph (lines 86–93) should be moved to the end of this section,

(2) the paragraph concerning the role of the IRE1α–XBP1 pathway in the regulation of ERAD (lines 104–126) should be placed immediately after the sentence (lines 95–98) beginning with “Once IRE1α is activated”,

(3) the same sentence should be shortened and read as follows: “Once IRE1α is activated, IRE1α causes the unconventional splicing of the 26-nucleotide intron of X-box binding protein 1 (XBP1), turning XBP1 into a transcription factor that regulates multiple pathways within the ER,” since the remaining part is too general and imprecise,

 (4) the fragment covering lines 99–103 should be placed after the paragraph describing the role of the IRE1α–XBP1 pathway in the regulation of ERAD,

(5) the sentences: „Under severe ER stress, IRE1α binds to TNF receptor associated factor 2 (TRAF2)41. This complex binds to apoptosis signal-regulating kinase (ASK1), which further activates c-Jun amino- terminal kinase (JNK) and cell apoptosis.” should be moved to last section, where information on Bax and Bak oligomerization are presented.

Section: ATF6

For the sake of clarity, the following changes are proposed in the section starting at line 170: „Dissociation of HSPA5 from ATF6 leads to a conformational change from an oligomeric to a monomeric form and subsequent activation of ATF6 through reduction by Protein Disulfide Isomerase 5 (PDI). This reduction results in the exposure of Golgi-localization signals (GLS1 and GLS2) on ATF6. Once mobilized to the Golgi, the transmembrane domain of ATF6 is cleaved by Site-1 and Site-2 proteases (S1P and S2P), allowing the cytosolic fragment of ATF6 to translocate to the nucleus. The degree of reduction of ATF6 correlates with its activation level: the more ATF6 is reduced, the better substrate it becomes for S1P, leading to more efficient cleavage and nuclear translocation. Some studies suggest that ATF6 becomes a suboptimal but acceptable substrate for S2P cleavage only after S1P removes approximately 250 amino acids from its luminal domain. This sequential cleavage and activation mechanism is unique to ATF6 within the unfolded protein response (UPR) pathway.”

In the second section of this chapter, the authors should continue the discussion on the role of ATF6 as a transcription factor and its involvement in the regulation of specific signaling pathways, as depicted in Figure 3. Subsequently, in this context, they should discuss whether ATF6 functions as a pro-survival or pro-apoptotic factor in cancer.

Section: PERK

In the sentence beginning with “Like IRE1α…” (line 211), the following fragment should be removed: “including nuclear factor erythroid 2-related factor 2 (NRF2), eukaryotic initiation factor 2α (eIF2α), forkhead box class O (FOXO), and diacylglycerol (DAG),” since this information is repeated in the subsequent paragraphs. Therefore, I suggest revising the sentence as follows:

“Like IRE1α, PERK oligomerizes upon activation and interacts with different partners⁶⁰,⁶¹, as clearly illustrated in Figure 4.”

Lines 215–236: In this section, the authors describe NRF2 as a transcriptional activator (line 218) and state that “NRF2 binds the antioxidant response element (ARE) to transcriptionally regulate various antioxidant enzymes.” (lines 223-224). However, at the beginning of this chapter, they wrote that NRF2 binds to and phosphorylates different proteins. Therefore, for the sake of clarity, the authors should carefully describe the role of NRF2 as a kinase and its interactions with other proteins, and only then address its function as a transcription factor. In addition, the information presented in this section does not correspond well with what is shown in Figure 4.

Lines 237–263: In this section, the regulatory role of PERK in protein translation through the phosphorylation of eIF2α is described. Considerable attention is also given to the role of the transcription factor ATF4 in the cellular stress response; however, there is a lack of information on the relationship between the increased expression of this protein and the phosphorylation of eIF2α — that is, the mechanistic link between PERK and ATF4 should be clarified.

Section: Non-Canonical Transcription Factors in UPR – CREB3 family

When describing the biological activities of CREB, for the sake of clarity, the authors should first focus on the canonical functions of CREB as a transcription factor: that is, its activities resulting from the binding of regulatory DNA sequences and only then discuss the activities arising from its interactions with other proteins.

Lines 311–312: There is no prior explanation of how HSPA5 reaches the Golgi apparatus. Please clarify this mechanism.

Line 386: HERP does not bind directly to ubiquitin; instead, it forms a complex with the ubiquitin ligase Hrd1, thereby enhancing the ubiquitylation and degradation of misfolded proteins.

Section: UPR and immune cells

Line 486: Does this refer to viral infections? If so, please add this information.

Line 504: Provide the full name of TAM.

Lines 485–516: The discussion of lipid metabolism changes in this section should be mentioned only briefly. The detailed information should be moved to the chapter “UPR and lipid metabolism.”

Section: UPR and lipid metabolism

Line 550: Explain why the binding of SREBP1 to the MRP1 gene promoter leads to chemoresistance.

Lines 553–555: Explain how IRE1α suppresses sterol regulatory element-binding transcription factor 1 (SREBF1), diacylglycerol O-acyltransferase 2 (DGAT2), and lipase E (LIPE). Additionally, the terms diacylglycerol and lipase should be written in lowercase.

Lines 555–556: Clarify what is meant by “DGAT2 is a direct RIDD target gene.”

Lines 560 and 562: Should the names SCD and SCD promoter be changed to SCD1 and SCD1 promoter for consistency?

Line 567: Change MUFA to MYFA.

Line 569: Revise the sentence “PERK is also activated by lipid stress to induce proliferation and promote lipogenesis.” to “PERK is also activated by lipid stress to promote lipogenesis,” since it is already known that PERK induces proliferation.

Author Response

Thank you for the detailed suggestions and feedback. We have addressed them and given details below:

“The numerous figures included in the work are very helpful in understanding the main text. However, what is presented in them does not always correspond well with their descriptions in the text. The authors should carefully compare the content of the figures with the information provided in the text. Also, for clarity, the figure legends should be expanded to include explanations of the most important abbreviations used. In some figures, the labels are too small and unreadable (Fig. 3 and 8)” -> Thank you for pointing this out! We have edited the figures accordingly.

Attention should be paid to ensure that gene names are always written in italics. -> We have made the changes

Specific comments

Section: IRE1 – Canonical signaling

For the sake of clarity:

(1) the paragraph (lines 86–93) should be moved to the end of this section -> Agree; we have moved it accordingly

(2) the paragraph concerning the role of the IRE1α–XBP1 pathway in the regulation of ERAD (lines 104–126) should be placed immediately after the sentence (lines 95–98) beginning with “Once IRE1α is activated”, ->We have moved the sentences accordingly

(3) the same sentence should be shortened and read as follows: “Once IRE1α is activated, IRE1α causes the unconventional splicing of the 26-nucleotide intron of X-box binding protein 1 (XBP1), turning XBP1 into a transcription factor that regulates multiple pathways within the ER,” since the remaining part is too general and imprecise -> Thank you for the suggestion - we have rewritten the sentence according to the reviewer’s suggestion.

 (4) the fragment covering lines 99–103 should be placed after the paragraph describing the role of the IRE1α–XBP1 pathway in the regulation of ERAD. ->We agree and have moved the fragment there.

(5) the sentences: „Under severe ER stress, IRE1α binds to TNF receptor associated factor 2 (TRAF2)41. This complex binds to apoptosis signal-regulating kinase (ASK1), which further activates c-Jun amino- terminal kinase (JNK) and cell apoptosis.” should be moved to last section, where information on Bax and Bak oligomerization are presented. -> We have moved the sentences accordingly.

Section: ATF6

For the sake of clarity, the following changes are proposed in the section starting at line 170: „Dissociation of HSPA5 from ATF6 leads to a conformational change from an oligomeric to a monomeric form and subsequent activation of ATF6 through reduction by Protein Disulfide Isomerase 5 (PDI). This reduction results in the exposure of Golgi-localization signals (GLS1 and GLS2) on ATF6. Once mobilized to the Golgi, the transmembrane domain of ATF6 is cleaved by Site-1 and Site-2 proteases (S1P and S2P), allowing the cytosolic fragment of ATF6 to translocate to the nucleus. The degree of reduction of ATF6 correlates with its activation level: the more ATF6 is reduced, the better substrate it becomes for S1P, leading to more efficient cleavage and nuclear translocation. Some studies suggest that ATF6 becomes a suboptimal but acceptable substrate for S2P cleavage only after S1P removes approximately 250 amino acids from its luminal domain. This sequential cleavage and activation mechanism is unique to ATF6 within the unfolded protein response (UPR) pathway.” ->Thank you for the very detailed suggestion. We have rewritten the sentence similar to what has been suggested.

In the second section of this chapter, the authors should continue the discussion on the role of ATF6 as a transcription factor and its involvement in the regulation of specific signaling pathways, as depicted in Figure 3. ->We have highlighted more explicitly and gone more into detail on ATF6’s role as a transcription factor.

Subsequently, in this context, they should discuss whether ATF6 functions as a pro-survival or pro-apoptotic factor in cancer.--> We have mentioned that ATF6 is both pro-survival and pro-apoptosis in different contexts as there are conflicting data in the literature. We have addressed this nuance now.

Section: PERK

In the sentence beginning with “Like IRE1α…” (line 211), the following fragment should be removed: “including nuclear factor erythroid 2-related factor 2 (NRF2), eukaryotic initiation factor 2α (eIF2α), forkhead box class O (FOXO), and diacylglycerol (DAG),” since this information is repeated in the subsequent paragraphs. Therefore, I suggest revising the sentence as follows:

“Like IRE1α, PERK oligomerizes upon activation and interacts with different partners⁶⁰,⁶¹, as clearly illustrated in Figure 4.”-> Thank you for the suggestion. We have done this.

Lines 215–236: In this section, the authors describe NRF2 as a transcriptional activator (line 218) and state that “NRF2 binds the antioxidant response element (ARE) to transcriptionally regulate various antioxidant enzymes.” (lines 223-224). However, at the beginning of this chapter, they wrote that NRF2 binds to and phosphorylates different proteins. Therefore, for the sake of clarity, the authors should carefully describe the role of NRF2 as a kinase and its interactions with other proteins, and only then address its function as a transcription factor. In addition, the information presented in this section does not correspond well with what is shown in Figure 4. -> We have now clearly explained and highlighted NRF2’s role as a transcription factor in different pathways first and foremost. An example would be “NRF2 binds the antioxidant response element (ARE) to transcriptionally regulate different antioxidant enzymes[65]”.

Lines 237–263: In this section, the regulatory role of PERK in protein translation through the phosphorylation of eIF2α is described. Considerable attention is also given to the role of the transcription factor ATF4 in the cellular stress response; however, there is a lack of information on the relationship between the increased expression of this protein and the phosphorylation of eIF2α — that is, the mechanistic link between PERK and ATF4 should be clarified. -> We agree with this comment and tried to rectify this by clearly saying “PERK-mediated eIF2a phosphorylation also triggers the transcriptional upregulation of ATF4[70] which has many functions.”

Section: Non-Canonical Transcription Factors in UPR – CREB3 family

When describing the biological activities of CREB, for the sake of clarity, the authors should first focus on the canonical functions of CREB as a transcription factor: that is, its activities resulting from the binding of regulatory DNA sequences and only then discuss the activities arising from its interactions with other proteins. ->We have adopted this and more explicitly described CREB as a transcription factor, such as “In cervical cancer, CREB3 has been shown to transcriptionally activate homocysteine-inducible ER-stress-inducible (HERP)” and “CREB3 activates and binds to a consensus DNA elements such as unfolded protein response element (UPRE) to promote transcription of different mRNAs[119]”

Lines 311–312: There is no prior explanation of how HSPA5 reaches the Golgi apparatus. Please clarify this mechanism. -> Thank you for pointing this out as we missed this. We have removed the Golgi apparatus section, rewriting the sentence into “The unbound HSPA5 is typically transported back to the ER by the transmembrane KDEL receptor (KDELR)[91] via coat protein complex I (COPI) vesicular transport, regulated by KDELR1[92,93].”

Line 386: HERP does not bind directly to ubiquitin; instead, it forms a complex with the ubiquitin ligase Hrd1, thereby enhancing the ubiquitylation and degradation of misfolded proteins. -> We appreciate the feedback and we have corrected this into “Herp’s promoter, a known ERAD ubiquitin-like ER-membrane protein to degrade proteins[118]”

Section: UPR and immune cells

Line 486: Does this refer to viral infections? If so, please add this information. -> This is a generic statement not just applicable to viral infections.

Line 504: Provide the full name of TAM. -> We have added the full name of TAM. “The IRE1α-XBP1 pathway also polarizes tumor-associated macrophages (TAMs) via macrophage inducible Ca2+-dependent lectin receptor (Mincle)-dependent ER stress in an immunosuppressive pro-tumorigenic manner. This pro-tumorigenic polarization is further reinforced by changes in lipid composition; IRE1α-XBP1 and IRE1α-STAT3 are both simultaneously activated to ensure TAM survival in favor of the tumor[148].”

Lines 485–516: The discussion of lipid metabolism changes in this section should be mentioned only briefly. The detailed information should be moved to the chapter “UPR and lipid metabolism.” -> We believe that this paragraph bridges the UPR, immune, and lipid metabolism angles so we want to end the UPR-immune angle here.

Section: UPR and lipid metabolism

Line 550: Explain why the binding of SREBP1 to the MRP1 gene promoter leads to chemoresistance.--> We further elaborated how SREBP1 increases expression of MRP1 and how MRP1 leads to chemoresistance. “, IRE1α drives c-myc activation. c-myc is shown to activate sterol regulatory element binding protein 1 (SREBP1), which then binds to multidrug resistance protein 1 (MRP1) promoter and increases MRP1 expression, leading to MRP1-mediated drug efflux and chemotherapy resistance[160]. “

Lines 553–555: Explain how IRE1α suppresses sterol regulatory element-binding transcription factor 1 (SREBF1)diacylglycerol O-acyltransferase 2 (DGAT2), and lipase E (LIPE). Additionally, the terms diacylglycerol and lipase should be written in lowercase. -> We have made the changes “In triple negative breast cancer, inhibition of IRE1α RNAse is shown to increase levels of triacylglycerols (TAGs) and polyunsaturated fatty acids and lower levels of diacylglycerols (DAG) while upregulating of sterol regulatory element-binding transcription factor 1 (SREBF1), Diacylglycerol O-acyltransferase 2 (DGAT2), and Lipase E (LIPE) genes, hinting at a pivotal role of IRE1α in suppressing TAG biosynthesis.”

Lines 555–556: Clarify what is meant by “DGAT2 is a direct RIDD target gene.” ->Thank you, we missed that this could be confusing. We have rewritten it as “IRE1α-mediated RIDD activity degrades DGAT2 mRNA”

Lines 560 and 562: Should the names SCD and SCD promoter be changed to SCD1 and SCD1 promoter for consistency? ->We have fixed this for consistency.

Line 567: Change MUFA to MYFA. ->We believe that monounsaturated fatty acids should be abbreviated as MUFA.

Line 569: Revise the sentence “PERK is also activated by lipid stress to induce proliferation and promote lipogenesis.” to “PERK is also activated by lipid stress to promote lipogenesis,” since it is already known that PERK induces proliferation. -> We have made this change.

Reviewer 2 Report

Comments and Suggestions for Authors

  P.M. Quan Mai, Tam-Anh Truong, Sai Kumar Samala, Bhoomika Muruvekere Lakshmisha, et al. 

The Unfolded Protein Response – Novel Mechanisms, Chal-2 lenges, And Key Considerations For Therapeutic Intervention   

Reviewer's comments:  

The submitted review is dedicated to the important subject and will be interesting for molecular oncologists. The manuscript is well written and nicely illustrated. There is a minor criticism: 

The Authors have mentioned ‘radiation therapy resistance’ among key words for this review paper; however, this point is not reviewed in their manuscript. The Authors should describe the role of UPR components/pathways in cancer cell radioresistance and discuss a possibility of tumor radiosensitization by targeting the UPR components/pathways (and provide with relevant refs, certainly). Perhaps, it would be nice to compile a separate, short subsection on UPR and radiotherapy similarly to what is dedicated to UPR and chemotherapy.

It is not clear why the Authors hope that inhibitors of the UPR will be tolerable by cancer patients. It seems likely that such agents will be extremely toxic because they will decrease the normal tissue/organ resistance to (i) physiological stresses (inflammation, ischemia, endotoxication etc); (ii) chemotherapy; (iii) radiotherapy; (iv) infections (inhibitors of the UPR are supposed to be immunosuppressors, don’t them?). The Authors should explain why they expect that the cytotoxic/sensitizing effects of UPR-targeting agents will largely be manifested toward malignancies. If there are some specific approaches aimed at the selectivity of UPR inhibitor action toward cancer cells, the Authors should discuss it.  

Author Response

The submitted review is dedicated to the important subject and will be interesting for molecular oncologists. The manuscript is well written and nicely illustrated. ->Thank you for the positive feedback – the encouragement is appreciated.

There is a minor criticism: 

The Authors have mentioned ‘radiation therapy resistance’ among key words for this review paper; however, this point is not reviewed in their manuscript. The Authors should describe the role of UPR components/pathways in cancer cell radioresistance and discuss a possibility of tumor radiosensitization by targeting the UPR components/pathways (and provide with relevant refs, certainly). Perhaps, it would be nice to compile a separate, short subsection on UPR and radiotherapy similarly to what is dedicated to UPR and chemotherapy. ->Thank you for the suggestion. We have removed radiation therapy resistance as part of our keywords. However, we have mentioned a few key points on how UPR influences radiation resistance: 1) NRF2 has been linked to radiation resistance “In colorectal cancer, irradiation, via KRAS activation, led to NRF2 upregulation which further enhanced 53-BP1 mediated non-homologous end-joining repair in malignant cells[194].” And “Also in esophageal squamous cell carcinoma, fractionated radiation is known to trigger ΔNP63α activation leading to increased NRF2 translocation to the nucleus, thus driving resistance to therapy[195]. Inhibitors of the NRF2-ARE pathway can therefore be employed as a complement to radiation induced cell death.” 2) Anti-HSPA5 therapies have been shown to sensitize tumors to radiation “Also, in glioblastoma, lung cancers and pancreatic cancer, radiation induced cs-HSPA5 upregulation has been recorded. In such cases, upon targeting with systemic monoclonal antibody therapies, cs-HSPA5 blockade was shown to enhance the efficacy of radiotherapy in tumor models[231,235]. Simultaneous and sequential cs-HSPA5 targeting in conjunction with irradiation, therefore presents an enticing one-two punch strategy for managing malignant tumors.”

It is not clear why the Authors hope that inhibitors of the UPR will be tolerable by cancer patients. It seems likely that such agents will be extremely toxic because they will decrease the normal tissue/organ resistance to (i) physiological stresses (inflammation, ischemia, endotoxication etc); (ii) chemotherapy; (iii) radiotherapy; (iv) infections (inhibitors of the UPR are supposed to be immunosuppressors, don’t them?). The Authors should explain why they expect that the cytotoxic/sensitizing effects of UPR-targeting agents will largely be manifested toward malignancies. If there are some specific approaches aimed at the selectivity of UPR inhibitor action toward cancer cells, the Authors should discuss it.  ->Thank you for the suggestion and we have made according changes. Though we did not explicitly say we expect UPR inhibitors are expected to be tolerable, we have suggested a few strategies. 1) Better patient stratification by targeting IRE1a in cancer patients who received chemotherapies. “Interestingly, up to 65% of patients who responded poorly to ER stress directed chemotherapeutics, demonstrated an overexpression of IRE1α[206]. Targeting IRE1α therefore is a promising strategy. Since the UPR is a ubiquitous cellular response to stress, tumor selective targeting of the IRE1α axis is unattainable unless inhibitors are selectively delivered /targeted against novel tumor antigens. However, amplification and overexpression of IRE1α in cancers offers a non-conventional chemotherapeutic target. Therefore, targeting IRE1α in such patient populations could potentially lead to therapeutic benefits with diminished systemic toxicity compared to conventional anti-cancer agents (taxanes, platinum compounds and anthracyclines).” 2) Target radiation-induced cell-surface HSPA5 “Interestingly, up to 65% of patients who responded poorly to ER stress directed chemotherapeutics, demonstrated an overexpression of IRE1α[206]. Targeting IRE1α therefore is a promising strategy. Since the UPR is a ubiquitous cellular response to stress, tumor selective targeting of the IRE1α axis is unattainable unless inhibitors are selectively delivered /targeted against novel tumor antigens. However, amplification and overexpression of IRE1α in cancers offers a non-conventional chemotherapeutic target. Therefore, targeting IRE1α in such patient populations could potentially lead to therapeutic benefits with diminished systemic toxicity compared to conventional anti-cancer agents (taxanes, platinum compounds and anthracyclines).” 3) Exploit nanoparticles’ intrinsic enhanced permeability and retention effects on tumors to trigger UPR-mediated cell death.

Reviewer 3 Report

Comments and Suggestions for Authors

Quan Mai et al present a narrative article focused on UPR signaling taking into account established information, possible interventions using common perturbagens, delivery approaches as well as future challenges that the field has to face to identify new clinical breakthroughs. In addition, the authors offered an interesting point of view about the possibility to target UPR in cancers, a theme of interest for a broad audience. The overall manuscript reads well and the cited literature is well positioned to support authors' statements. Hence, the manuscript results readable and understandable also by those that are not expert in the field. However, over UPR activation, proteotoxic stress can trigger ER-phagy. Such a condition is also experience by cancer cells as a salvage pathway to survive to hypoxia-induced ER stress. What is currently known is that the interaction between BiP and FAM134B seems to be crucial, thus tightly linking UPR and ER-phagy pathways. For this, and other reasons established in literature, this referee considers important to expand or integrate the content of this review by introducing ER-phagy and its relation(s) with UPR. 

Author Response

Quan Mai et al present a narrative article focused on UPR signaling taking into account established information, possible interventions using common perturbagens, delivery approaches as well as future challenges that the field has to face to identify new clinical breakthroughs. In addition, the authors offered an interesting point of view about the possibility to target UPR in cancers, a theme of interest for a broad audience. The overall manuscript reads well and the cited literature is well positioned to support authors' statements. Hence, the manuscript results readable and understandable also by those that are not expert in the field. ->Thank you for the positive feedback!

However, over UPR activation, proteotoxic stress can trigger ER-phagy. Such a condition is also experience by cancer cells as a salvage pathway to survive to hypoxia-induced ER stress. What is currently known is that the interaction between BiP and FAM134B seems to be crucial, thus tightly linking UPR and ER-phagy pathways. For this, and other reasons established in literature, this referee considers important to expand or integrate the content of this review by introducing ER-phagy and its relation(s) with UPR.  We missed the importance of ER-phagy on UPR in cancer and have added a new section! We have also emphasized the interaction between BiP and FAM134B.

Reviewer 4 Report

Comments and Suggestions for Authors

The authors in this review discussed the detailed molecular mechanisms of the Unfolded Protein Response (UPR) in cancer. They covered key interactions among IRE1, PERK, ATF6, and HSPA5, as well as the non-canonical roles of the UPR in immune regulation and lipid metabolism, and their dual nature in cancer cells. The manuscript is well written and timely, discussing emerging UPR inhibitor therapies, challenges in targeting the UPR, and ongoing clinical trials. The authors noted that recent clinical trials of PERK and HSPA5 inhibitors have shown limited benefits. Adding the authors' perspectives on failed trials, toxicity issues, and strategies to improve the outcomes would strengthen the manuscript.

Minor text edits: inconsistent use of text, for example, “anti-tumor” (line 726) and “antitumor” (line 705); missing periods at the end of the abstract and Fig. 3.

Author Response

The authors in this review discussed the detailed molecular mechanisms of the Unfolded Protein Response (UPR) in cancer. They covered key interactions among IRE1, PERK, ATF6, and HSPA5, as well as the non-canonical roles of the UPR in immune regulation and lipid metabolism, and their dual nature in cancer cells. The manuscript is well written and timely, discussing emerging UPR inhibitor therapies, challenges in targeting the UPR, and ongoing clinical trials.  -> Thank you for the encouragement and the positive feedback is truly appreciated!

The authors noted that recent clinical trials of PERK and HSPA5 inhibitors have shown limited benefits. Adding the authors' perspectives on failed trials, toxicity issues, and strategies to improve the outcomes would strengthen the manuscript. ->We have attempted to add our opinions in our clinical trial reports. However, the trials either are ongoing or have been terminated without clear indication why.

Minor text edits: inconsistent use of text, for example, “anti-tumor” (line 726) and “antitumor” (line 705); missing periods at the end of the abstract and Fig. 3. -> Thank you for pointing this out and we have made the changes!